



# Resolving the effects of 2-D versus 3-D grain measurements on apatite (U–Th) / He age data and reproducibility

**Emily H. G. Cooperdock**[1,2,a]**, Richard A. Ketcham**[1]**, and Daniel F. Stockli**[1]

[1]Department of Geological Sciences, University of Texas at Austin, Austin, 78712, USA
[2]Woods Hole Oceanographic Institution, Woods Hole, 02543, USA
[a]now at: Department of Earth Sciences, University of Southern California, Los Angeles, CA 90089, USA

**Correspondence:** Emily H. G. Cooperdock (cooperdo@usc.edu)

Received: 9 May 2019 – Discussion started: 27 May 2019
Accepted: 26 August 2019 – Published:

**Abstract.** (U–Th) / He thermochronometry relies on the accurate and precise quantification of individual grain volume and surface area, which are used to calculate mass, alpha ejection ($F_T$) correction, equivalent sphere radius (ESR), and ultimately isotope concentrations and age. The vast majority of studies use 2-D or 3-D microscope dimension measurements and an idealized grain shape to calculate these parameters, and a long-standing question is how much uncertainty these assumptions contribute to observed intra-sample age dispersion and accuracy. Here we compare the results for volume, surface area, grain mass, ESR, and $F_T$ correction derived from 2-D microscope and 3-D X-ray computed tomography (CT) length and width data for > 100 apatite grains. We analyzed apatite grains from two samples that exhibited a variety of crystal habits, some with inclusions. We also present 83 new apatite (U–Th) / He ages to assess the influence of 2-D versus 3-D $F_T$ correction on sample age precision and effective uranium (eU). The data illustrate that the 2-D approach systematically overestimates grain volumes and surface areas by 20 %–25 %, impacting the estimates for mass, eU, and ESR – important parameters with implications for interpreting age scatter and inverse modeling. $F_T$ factors calculated from 2-D and 3-D measurements differ by ∼ 2 %. This variation, however, has effectively no impact on reducing intra-sample age reproducibility, even on small aliquot samples (e.g., four grains). We also present a grain-mounting procedure for X-ray CT scanning that can allow hundreds of grains to be scanned in a single session and new software capabilities for 3-D $F_T$ and $F_T$-based ESR calculations that are robust for relatively low-resolution CT data, which together enable efficient and cost-effective CT-based characterization.

## 1 Introduction

(U–Th) / He thermochronometry of accessory phases, such as apatite and zircon, has been widely applied to study tectonic, volcanic, and surface processes (e.g., Zeitler et al., 1987; Stockli et al., 2000; Ehlers and Farley, 2003; Reiners and Brandon, 2006). The method is based on the radiogenic accumulation of He from the alpha decay of U, Th, and Sm isotopes and the diffusive loss of He via thermal processes. In addition, He is lost due to the "long alpha stopping distances" associated with the kinetic energy of alpha decay (∼ 5 MeV), requiring a shape-based alpha ejection correction ($F_T$ correction) (Farley et al., 1996). This correction as traditionally applied includes several simplifications and assumptions, such as an idealized grain geometry and homogenous parent nuclide concentrations (Farley et al., 1996; Farley, 2002; Ketcham et al., 2011). It has been shown that due to uncertainties in grain geometry, stopping distances, and parent nuclide zonation and variability, this correction can contribute > 50 % of the total analytical uncertainty (Farley and Stockli, 2002). Similarly, low error, highly dispersed apatite (U–Th) / He ages are problematic for robust interpretation and time–temperature modeling (e.g., Fox et al., 2019). The observation that the scatter of measured ages in even well-understood samples exceeds expectation based on analytical errors, combined with the knowledge that the above simplifications will not always hold, has led to the practice of

reporting errors derived from the reproducibility of standards rather than propagated analytical uncertainties in He dating. While the effect and mitigation of parent nuclide zonation in apatite and zircon to improve the accuracy and precision of (U–Th) / He ages have been studied (e.g., Farley et al., 1996; Hourigan et al. 2005; Ketcham et al. 2011; Gautheron et al., 2012; Bargnesi et al., 2016; Danisik et al., 2017; McDannell et al., 2018), the effects of grain morphology and measurement on age, uncertainty, and intra-sample variability are less known, with only a few previous studies on improvements to grain measurement (Herman et al., 2007; Evans et al., 2008; Glotzbach et al., 2019).

In practice, for the determination of a correct He age, the grain dimensions and shape must be measured to compute an $F_T$ correction factor prior to He and U, Th, and Sm analysis, assuming either parent nuclide homogeneity or prescribing an assumed or measured 1-D or 2-D parent nuclide zonation (Farley et al., 1996; Farley, 2002). While not directly related to the computation of He ages, these same grain dimensions are also used to calculate grain size parameters for the purpose of calculating isotopic and/or elemental concentrations and for age interpretation and diffusion or thermal history modeling (Shuster et al., 2006; Flowers et al., 2007, 2009; Flowers, 2009; Gautheron et al., 2009; Flowers and Kelley, 2011). For example, the grain mass, which is used to calculate the grain U, Th, Sm, and He concentrations, is often derived from the grain volume and an assumed density. Similarly, correlation between grain size (ESR) and He aliquot age has been used for qualitative and quantitative thermal history reconstruction using He diffusivity models (Reiners and Farley, 2001; Flowers and Kelley, 2011). Thus, the ability to measure accurate and precise grain dimensions, volumes, and surface areas for mineral grains has cascading effects for the determination, reporting, and interpretation of (U–Th) / He data.

Most commonly, $F_T$, volume, and surface area are calculated using two or three grain dimensions (length + width 1± width 2) measured in 2-D on an optical microscope using imaging software with a micrometer-based calibration. This approach requires the assumption of an idealized grain shape that most closely matches the mineral habit, such as a hexagonal prism for apatite or tetragonal prism for zircon, while simplifying (or ignoring) the more complex grain terminations (Farley et al., 1996; Farley, 2002). Hence, it has been best practice to select euhedral mineral grains to most closely match assumed, idealized grain shapes and large grains to minimize the amplification of uncertainties related to the $F_T$ correction. However, even in felsic magmatic samples with high-quality apatite, grains are often characterized by a wide range of grain shapes, variations in grain terminations, and the potential for broken or chipped surfaces that cause deviations from the idealized hexagonal prism. Furthermore, apatite grains often do not represent symmetric or equidimensional hexagonal prisms and are characterized by varying face widths, commonly, but also possibly inconsistently,

lying on their largest and flattest face on the microscope slide and thus potentially introducing systematic biases during the selection of the clearest, inclusion-free grains.

Recognizing that this optical-microscopy approach is both limiting and may be an important source for error or bias in (U–Th) / He ages and their interpretation, more sophisticated approaches have been proposed to determine grain dimensions, namely methods that do not require assuming a grain shape (Herman et al., 2007; Evans et al., 2008; Glotzbach et al., 2019). One approach presented by Glotzbach and others (2019), called "3DHe", is an openly available software that uses orthogonal 2-D grain photos to model accurate 3-D grain shapes. Another approach is to employ X-ray computed tomography (CT) to determine accurate grain shapes in an effort to improve precision and accuracy in $F_T$ and (U–Th) / He age determinations (Herman et al., 2007; Evans et al., 2008; Glotzbach et al., 2019). Herman et al. (2007) used 3-D CT grain dimensions to calculate $F_T$ factors and present a production–diffusion model to extract thermal histories for detrital apatite grains. Evans et al. (2008) and Glotzbach et al. (2019) both tested the efficacy of 2-D microscope measurements against 3-D CT data of zircon and apatite grain shape and size, arriving at quite different estimated discrepancies between microscope measurements and the CT data (1 %–24 % and < 1 %–6 %, respectively).

This new study investigates the effect of 2-D versus 3-D grain geometry measurement techniques on grain dimension, volume, surface area, ESR, mass, $F_T$, and the corrected age as well as effective uranium (eU) concentrations. In contrast to previous studies, which used 5–24 grains, we characterized > 100 apatite grains from two granitic samples for a more statistically robust comparison and in an effort to more systematically capture variations in apatite morphologies and sizes, as well as to screen for inclusions. We chose samples from crystalline basement that experienced fast-cooling histories in order to target the impact of grain measurement techniques and minimize the effects of cooling history and transport on the (U–Th) / He age and dispersion. The apatite grains were picked and measured by a single analyst using 2-D optical techniques and then CT scanned. Building on previous work, we present a method for relatively rapid scans of > 100 grains at 4–5 μm resolution, enabling affordable and efficient 3-D screening. We introduce the capabilities of an updated version of Blob3D (Ketcham, 2005; freely distributed software) that allows for the efficient batch processing of CT-scanned grains and outputs parameters such as grain volume and 3-D $F_T$. We further develop an approach for calculating ESR on the basis of equivalent $F_T$ rather than an equivalent surface-to-volume ratio as a more direct and accurate means of approximating the diffusional domain as a sphere. Finally, in contrast to previous studies, we use the results of > 80 apatite (U–Th) / He ages to evaluate the reliability of the 2-D measurements as well as the impact on the (U–Th) / He age and uncertainty.

## Geologic background of the samples

For this study, we selected two plutonic samples from the Cretaceous Cordilleran magmatic arc in the western USA that yielded abundant, high-quality apatite and have been part of previous thermochronometric studies. Sample 97BS-CR8 is from a granodiorite in the Carson Range in the eastern Sierra Nevada along the Nevada–California border. The sample yielded an apatite fission track age of $68 \pm 2$ Ma ($P(X^2) =$ 75.4 %, 25 grains, $N_s = 1341$) (Surpless et al., 2002). The second sample, 95BS-11.3, is from a quartz monzonite exposed in the Wassuk Range in western Nevada, exhumed during Basin and Range normal faulting. The sample has a reported apatite fission track age of $16.3 \pm 1.4$ Ma ($P(X^2) =$ 76.1 %, 30 grains, $N_s = 158$) and apatite (U–Th) / He age of $9.9 \pm 1.9$ Ma (Stockli et al., 2002). These samples were chosen for their abundant apatite and relatively simple cooling histories. Their geologic histories are relevant to the present study in that the apatite grains derive from plutonic rocks and did not experience complex metamorphic or magmatic histories, nor natural abrasion during sedimentary transport. Furthermore, both are plutonic samples that experienced rapid post-magmatic cooling or fault-related exhumation and are expected to have spent little time in the apatite He partial retention zone.

## 2 Methods

### 2.1 Grain selection and 2-D measurements

Apatite grains were picked from two samples, 97BS-CR8 ($n = 50$) and BS95-11.3 ($n = 62$), using a Nikon SMZ-U/100 optical microscope at a total magnification of $180 \times$. Apatite grains were selected to include the range of grain morphologies present in the sample (e.g., broken, flat, and prismatic ends). Intentionally, several grains with visible inclusions were also selected to evaluate how well these inclusions showed up in the CT scans. All apatite grains were photographed using a Nikon digital ColorView camera connected to the microscope. The short and long axes were measured manually using AnalySIS® imaging software (Figs. 1 and 3). We chose to measure a single width and did not flip the apatite 90° because this is still common practice in many labs and would allow us to compare the "simplest" 2-D measurement approach with the 3-D CT data. For sample BS95-11.3, grains were imaged and measured on double-sided sticky tape in preparation for the CT mount (Fig. 1). However, we determined that this can cause grains to sit in upright orientations, which is fine for CT scanning but not for 2-D measurements. For sample 97BS-CR8 each apatite grain was placed on a glass slide for 2-D measurements and then transferred to the sticky tape for the CT mount to remedy this issue (Fig. 1).

### 2.2 Grain-mounting procedure for CT

Once the grains were measured optically in 2-D, they were mounted for CT scanning by orienting several tens of grains on a plastic disk and stacking multiple disks (Fig. 2). The procedure to create a single-layer mount for multi-grain scanning entails covering a flat top of a pushpin with double-sided sticky tape that can be precut using a standard hole punch. Apatite grains are then picked directly onto the tape in a grid-like pattern. The pushpin surface is $\sim 5$ mm in diameter, which easily allows $\geq 50$ apatite grains to be mounted in one layer, tightly spaced, without touching. Grains could be packed more densely as long as they can be reliably identified after scanning; they can even be touching, although this leads to a small increase in processing time to separate them using functions in the Blob3D software.

To utilize the total scanned volume, at least five multi-grain layers can be stacked for a single scan (up to 5 mm tall). To create stackable layers, sturdy plastic disks are made using a standard hole punch, with one side of the disk covered with double-sided sticky tape and apatite grains mounted in the procedure outlined above. Once all the layers are mounted and all excess tape is trimmed, the disks are stacked on top of the push pin. The arrangement is secured by a thin wrap of parafilm. The parafilm and sticky tape are critical to ensure the crystals and layers do not move during scanning. This mount can be easily disassembled after scanning to retrieve the grains for further analysis.

### 2.3 X-ray CT scanning

The multi-grain mounts were scanned with a Zeiss Xradia MicroXCT scanner at the University of Texas High-Resolution X-ray CT Facility (Ketcham and Cooperdock, 2019). Optimal scanning parameters will vary with the instrument being used, with top priorities being to minimize scanning artifacts and noise, while also minimizing time and cost. Lower X-ray energies are more sensitive to compositional variations but more prone to beam-hardening artifacts. We experimented with various settings in this study. The grain mount for sample 97BS-CR8 was scanned with X-rays set at 100 kV and 10 W, with a 1.0 mm $SiO_2$ filter. 1153 views were gathered at 1.5 s per view, for an acquisition time of 28.9 min. Source–mount distance was 37.7 mm, and mount–detector distance was 12.8 mm. The $2048 \times 2048$ camera data were binned by two, and the lower-energy X-rays and weaker filtering necessitated the application of a beam-hardening correction during reconstruction. The reconstructed data had a voxel (3-D pixel) size of 5.03 μm.

The grain mount for sample BS95-11.3 was scanned with X-rays set at 150 kV and 10 W with a 1.6 mm $CaF_2$ beam filter, acquiring 571 views at 1.5 s per view, for an acquisition time of 14.3 min, not including calibration. Source–mount distance was 37.7 mm, and mount–detector distance was 17.8 mm. The camera data were binned by 2, and no

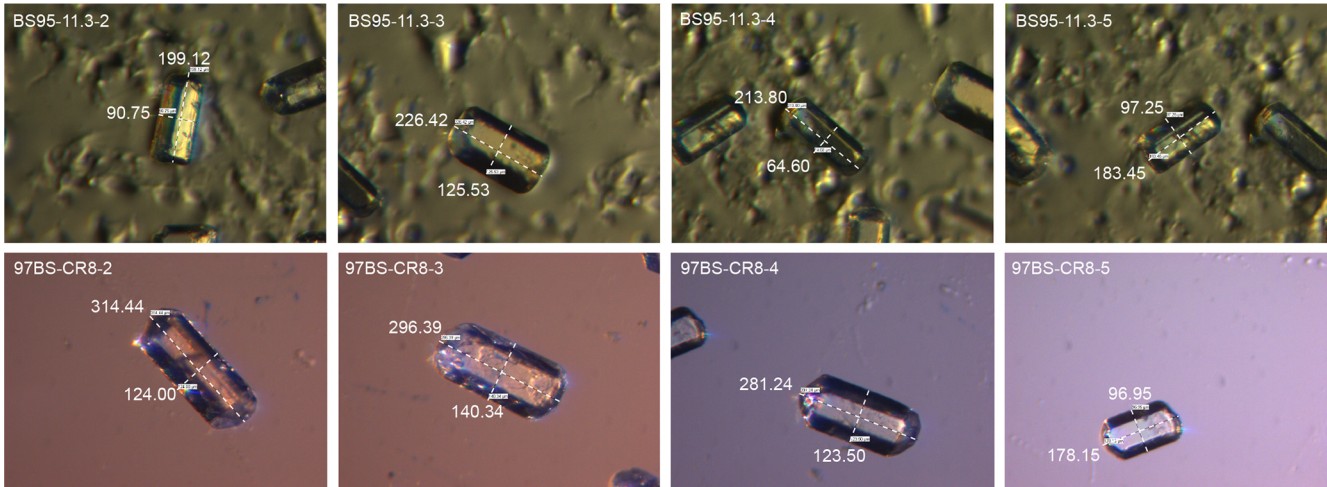

**Figure 1.** Apatite grain photos with 2-D measurements taken on an optical microscope. Dimensions are reported in micrometers and the grain aliquot name is in the top left corner of each photo. The top row is photographed on double-sided sticky tape, and the bottom row is photographed on a glass slide.

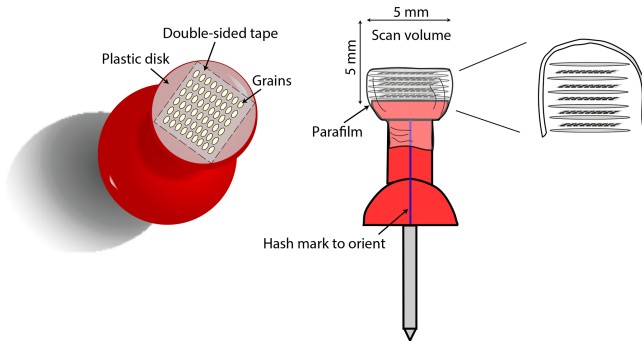

**Figure 2.** Schematic rendering of the CT-mounting procedure. Grains are adhered to the top of a plastic disk using double-sided sticky tape, with multiple grains placed onto a 5 × 5 mm surface. Multiple plastic disk layers with grains may be assembled and then stacked to take full advantage of the height of the scan. These layers are held together using parafilm, and a hash mark on the pushpin enables further orientation of the scan in order to retrieve the grains afterwards for further analysis.

beam-hardening correction was applied during reconstruction. The resulting data had a voxel size of 4.58 μm.

Example images from the two datasets are shown in Fig. 3, illustrating some of the trade-offs. The scan data for BS95 are noisier, primarily due to the faster acquisition, higher X-ray energy, and more severe filtering. Even with this level of noise, high-attenuation inclusions are evident. The scan data for 97BS are less noisy, allowing for the detection of a fluid inclusion, but beam hardening due to the lower-energy X-ray spectrum has caused faint streaks to emanate from or connect some grains. These subtle artifacts have a negligible effect on measurements but may be expected to increase in severity with more or higher-density grains.

## 2.4 Grain size and shape, $F_T$, mass calculations

### 2.4.1 2-D measurement calculations

The microscope length and width measurements are used to calculate volume and surface area, which are then used to calculate mass, ESR, and $F_{T,U}$ and $F_{T,\text{Th}}$ for each apatite grain, following methods laid out in Farley et al. (1996), Farley (2002), and Farley and Stockli (2002) (Fig. 4). An equidimensional hexagonal prism geometry was assumed with the length ($L$) measurement for height of the prism and the half-width ($r$) for the radius of the prism. All equations used for calculating these parameters are included below or in the Appendix.

Volume ($V$):

$$V = \frac{3 \times \sqrt{3}}{2} \times r^2 \times L, \tag{1}$$

where $L$ is height and $r$ is the half-width.

Surface area (SA):

$$\text{SA} = 6 \times r \times L + 3\sqrt{3} \times r^2, \tag{2}$$

where $L$ is height and $r$ is the half-width.

Equivalent spherical radius (ESR):

$$\text{ESR} = 3 \times \frac{V}{\text{SA}}. \tag{3}$$

Mass:

$$\text{mass} = 3.2 \left(\frac{g}{\text{cm}^3}\right) \times V(\text{mm}^3) \times 1000. \tag{4}$$

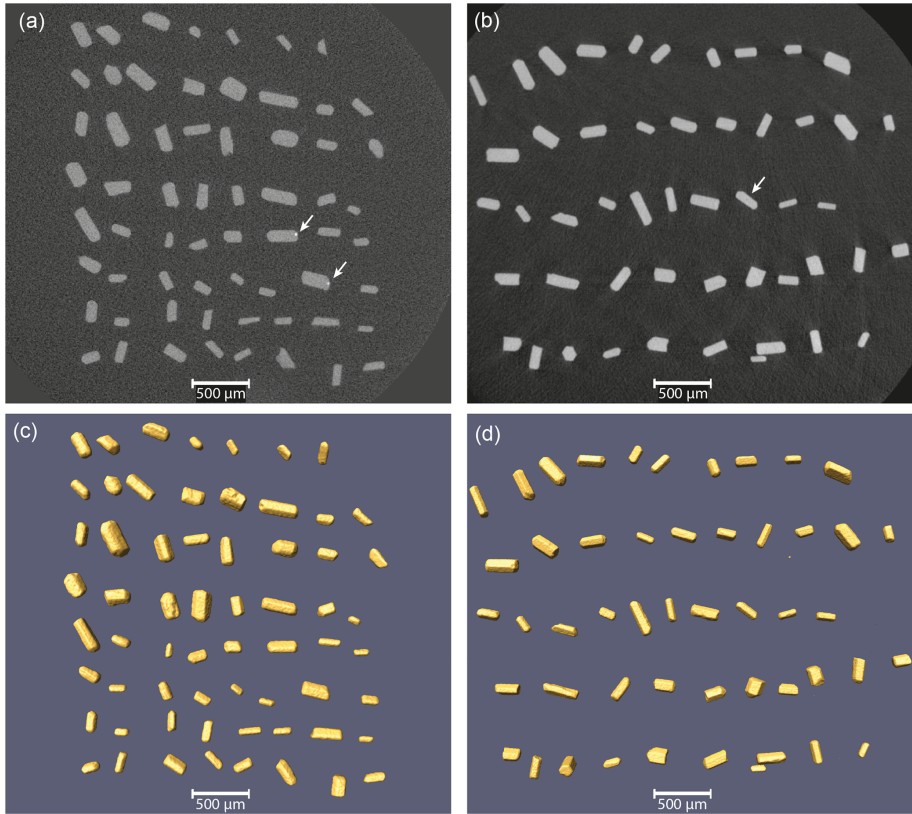

**Figure 3.** Example CT slices (**a, b**) and 3-D renderings (**c, d**) of apatite grain mounts for BS95 (**a, c**) and 97BS (**b, d**). Arrows indicate two grains with high-attenuation mineral inclusions in BS95 and a fluid inclusion in 97BS. The CT slice for 97BS is actually an oblique slice through the original data to allow all grains to appear in the same image.

$F_{T,U}$ and $F_{T,\mathrm{Th}}$ (2-D case; e.g., Farley, 2002):

$$F_{T,U} = 1 - \left(5.13 \times \frac{\mathrm{SA}}{V}\right) + \left(6.78 \times \frac{\mathrm{SA}^2}{V}\right),$$

$$F_{T,\mathrm{Th}} = 1 - \left(5.9 \times \frac{\mathrm{SA}}{V}\right) + \left(8.99 \times \frac{\mathrm{SA}^2}{V}\right). \tag{5}$$

Mean $F_T$ (see Appendix for explanation) from Farley et al. (1996) (used here for 2-D calculations):

$$F_T = a_{238} \times F_{T,U} + (1 - a_{238}) \times F_{T,\mathrm{Th}}, \tag{6a}$$

where $a_{238} = \left(1.04 + 0.245 \times \frac{\mathrm{Th}}{\mathrm{U}}\right)^{-1}$.

From Blob3D for this study (used here for 3-D calculations):

$$\overline{F_T} = A_{238} F_{T,238} + A_{232} F_{T,232} + (1 - A_{238} - A_{232})$$
$$F_{T,235}, \tag{6b}$$

where $A_{238} = \left(1.04 + 0.247 \left[\frac{\mathrm{Th}}{\mathrm{U}}\right]\right)^{-1}$ and $A_{232} = \left(1 + 4.21 / \left[\frac{\mathrm{Th}}{\mathrm{U}}\right]\right)^{-1}$.

Effective uranium concentration (eU) (see Appendix for explanation):

$$\mathrm{eU} = [U] + 0.238 [\mathrm{Th}] + 0.0012 [\mathrm{Sm}]. \tag{7}$$

### 2.4.2 3-D calculations

Our principal 3-D calculations were implemented in Blob3D (Ketcham, 2005), a program written in the IDL programming environment for efficient measurement of the dimensions, shape, and orientation of discrete features in volumetric datasets. The typical Blob3D method for calculating volume is to segment the grains based on a threshold set at 50 % of the CT number (grayscale) difference between apatite and the surrounding air. If grains are touching, or close enough to touching that their selected regions are connected, the software provides several separation methods, the simplest being an erode–dilate procedure. Volume is calculated as the number of voxels in a grain multiplied by the voxel volume, and surface area is calculated by summing the areas of the triangular facets of an isosurface surrounding the grain, which is smoothed to reduce excess roughness from the cubic voxel edges. The shape parameters BoxA, BoxB,

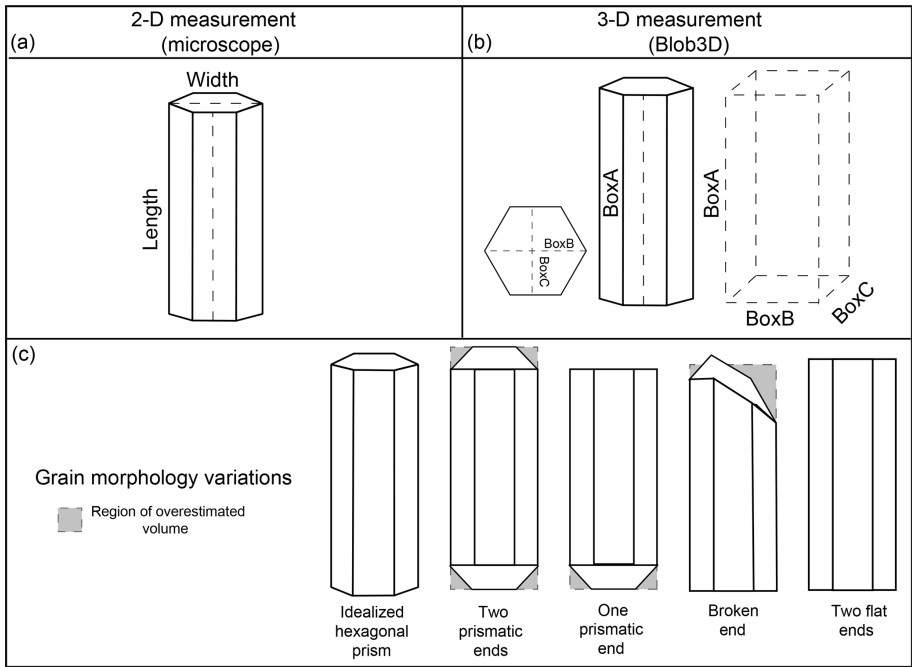

**Figure 4. (a, b)** Rendering of dimension data collected by 2-D and 3-D methods. Length and width are measured in 2-D using an optical microscope measuring the long and wide axis of a grain. Blob3D reports the length, width, and height (BoxA, BoxB, and BoxC) based on the best-fit rectangle for the grain dimensions. **(c)** Rendering of the full range of variations in grain terminations exhibited by the apatite in this study. Highlighted in gray are potential areas of overestimated volume if an ideal hexagonal prism is assumed and calculated with 2-D length and width data.

and BoxC are respectively the length ($L$), width ($W$), and height corresponding to the dimensions of the smallest rectangular box that will enclose the grain (Ketcham and Mote, 2019). BoxC is calculated as the shortest 3-D caliper length, BoxB is the shortest caliper length orthogonal to BoxC, and BoxA is the caliper length perpendicular to BoxC and BoxB (Fig. 4; Appendix C).

A Monte Carlo method was implemented to measure $F_T$, probably similar in many, but not all, respects to previous work (Herman et al., 2007; Glotzbach et al., 2019). Stopping distances for $^{238}$U, $^{235}$U, $^{232}$Th, and $^{147}$Sm for the set of minerals reported in Ketcham et al. (2011) are included in the software. Taking the set of selected voxels for a grain, the origin point for each alpha particle is selected by first randomizing from which voxel to start and then randomizing an $(x, y, z)$ location within that voxel. The direction for each particle is obtained by sequentially stepping through a list of near-uniformly distributed orientations calculated by starting with an octahedron and subdividing each triangular face four times until there are 1026 vertices, which are then scaled to lie on a unit sphere (Ketcham and Ryan, 2004). This approach provides slightly better precision than randomizing orientations, and 200 000 Monte Carlo samples are sufficient to get precision to within 0.1 % in all tests reported below. Separate $F_T$ factors for each decay chain ($F_{T,238}$, $F_{T,235}$, $F_{T,232}$, $F_{T,147}$) are calculated, and a revised method for cal-

culating mean $F_T$ that more precisely accounts for $^{235}$U is provided in Eq. (6) (explanation in Appendix A).

If the resolution of the scan is low with respect to the stopping distance ($^{238}$U stopping distance / voxel size < 4), excess surface roughness effects from voxelation are reduced by super-sampling. The voxels for each grain and the surrounding voxels are subdivided into 27 ($3^3$) elements, and the super-sampled volume is smoothed with a 5-voxel-wide cubic kernel. The result is then thresholded using a value that maintains the original volume as closely as possible.

These methods were tested on ideal spheres and cylinders, with radii of 63 and 31.5 μm and the latter with an aspect ratio of 4 (Appendix B). At voxel sizes up to 8 and 4 μm for the respective radii, mean $F_{T,238}$ values averaged within 0.2 % of the ideal-shape values for spheres; further doubling the voxel sizes raised the mean error to 0.5 %. Cylinders performed better, with a mean error of 0.3 % when voxel sizes were 1/4 of the radius.

In their Monte Carlo $F_T$ implementation, Herman et al. (2007) report poor precision for small spheres when their centers are not centered in a voxel, with errors rising to several percent for a 40 μm radius sphere with 6.3 μm voxels across a range of center locations (calculated $F_T$ range ∼ 0.58–0.67). Errors of this magnitude correspond to the effect of getting the radius wrong by plus or minus almost an entire voxel (∼ 15 % of the radius), too large to be reasonable

and probably caused by a problem with their test. We tested our segmentation method by running 100 000 trials randomizing the location of the sphere center using the same radius and voxel size and got maximum radius errors of $+0.8$ and $-1.1\%$ and a standard deviation of $0.2\%$ (Appendix B). We are thus confident that our implementation provides a high degree of accuracy and precision on even very small grains at low resolutions with voxel sizes up to $25\%$ of the radius.

We took three approaches to calculating ESR from the 3-D data. The first two are based on the equivalent surface-to-volume ratio (SV) approach (Meesters and Dunai, 2002). The model-based value $ESR_{SVm}$ uses the BoxA and BoxB caliper dimensions as $L$ and $W$ for Eqs. (1) through (3), while the 3-D CT-based value $ESR_{SV3D}$ uses the 3-D-measured volume and surface area for Eq. (3). Because of the unsupported assumptions of the model approach and the shortcomings of surface area measurements, both discussed below, neither of these solutions is ideal. An alternative ESR is based on the equivalent $F_T$ approach; Ketcham et al. (2011) demonstrated than an equivalent $F_T$ sphere provides a more accurate conversion for diffusion calculations than an equivalent SV one. The set of calculations to determine the $F_T$-equivalent sphere radius $ESR_{F_T}$ are provided in Appendix A.

## 2.5  (U–Th) / He procedure

The apatite (U–Th) / He ages were analyzed in the UTChron Thermochronology Laboratory at the University of Texas at Austin. Individual grains were measured, wrapped into platinum tubes, loaded into a 42-hole sample holder, and pumped to ultrahigh vacuum. Each aliquot was heated to $\sim 1070\,°C$ for 5 min using a Fusions Diode laser system. The released gas was spiked with a $^3$He tracer and purified by a Janis cryogenic cold trap at 40 K and SAES NP-10 getter prior to measurement of the $^4$He/$^3$He on a Blazers Prisma QMS-200 quadrupole mass spectrometer. Final $^4$He contents were calculated using a manometrically calibrated $^4$He standard of known concentration measured during the analytical run. All apatite aliquots were reheated once under the same conditions to ensure full gas release.

After degassing, the platinum packets containing the apatite grains were placed into plastic vials and dissolved in a 100 µL 30 % $HNO_3$ $^{235}U-^{230}Th-^{149}Sm$ spike solution for 90 min at 90 °C. After acid digestion, 500 µL of Milli-Q ultrapure $H_2O$ was added to dilute the solutions to $\sim 5\%$ $HNO_3$ and equilibrated for $\geq 24\,h$ prior to analysis. The solutions were analyzed using a Thermo Element2 high-resolution inductively coupled plasma–mass spectrometer (HR-ICP-MS) equipped with a 50 µL min$^{-1}$ microconcentric nebulizer. Final $^{238}U$, $^{232}Th$, and $^{147}Sm$ values were blank-corrected and calibrated using a spiked, gravimetrically calibrated $\sim 1\,ppb$ standard solution. Final (uncorrected) ages were calculated by solving the He age equation by means of Taylor series expansion and reported with a 6 % standard error based on long-term intra-laboratory analysis of apatite age standards. Corrected final ages are determined by dividing the uncorrected age by the mean $F_T$ factor (Eq. 5). U, Th, and Sm concentrations, although not used in the age calculations, were determined for reporting purposes using the grain volumes and a nominal apatite density (e.g., Fig. 4, Eq. 4).

## 3  Results

CT scanning combined with Blob3D analysis provides 3-D grain-specific volume, surface area, dimensions, and $F_T$ factors for each decay chain. The 2-D optical measurements provide dimension information, which is used to calculate volume, surface area, $F_{T,U}$, and $F_{T,Th}$ based on an assumed grain geometry of an equidimensional hexagonal prism (all results are reported in the Appendix). We assume that the 3-D-measured volume and $F_T$ values are sufficiently accurate to benchmark the 2-D data (all comparisons reported in Table 1 and Fig. 5). Surface area is more problematic to benchmark due to a number of factors, such as fractal roughness, CT data blurring, and voxelation effects, as discussed below, and thus 2-D and 3-D results can only be compared in a relative sense for surface area.

2-D and 3-D data are compared for each sample and as an entire population in Tables 1 and 2. The average 3-D / 2-D ratio of each parameter is reported with its $1\sigma$ standard deviation. This average ratio shows whether the 2-D measurements on average overestimate (ratio $< 1$) or underestimate (ratio $> 1$) the 3-D measurements. Also reported is the absolute percent difference between the 2-D and 3-D measurements to illustrate the magnitude of deviation between the measurements. While comparing 2-D and 3-D results, it became apparent that one 2-D grain measurement was made at an incorrect microscope magnification setting, causing the length and width to be off by 2 times, far greater than every other grain measured. Hence, this grain measurement (97BS-CR8-1) was not included when calculating the average differences between 3-D and 2-D measuring techniques.

## 3.1  Grain factors

Grains from both samples display a range of habits typical for apatite, including two flat ends, two prismatic ends, one flat and one prismatic end, and one or two broken or chipped ends (Figs. 1 and 4). The grain morphology and the presence of any visible inclusions were recorded during hand-picking (Table 2). Surprisingly, there are no clear systematic relationships between the presence of inclusions and grain age or grain shape and ESR, volume, or surface area. The 2-D length measurements are on average $\sim 2\%$ smaller than the 3-D BoxA dimension. On the other hand, the 2-D width dimension is on average $\sim 3\%$ greater than the 3-D BoxB dimension (Table 1).

One inevitable source of uncertainty in 2-D length and width measurements is analyst judgment and error. For ex-

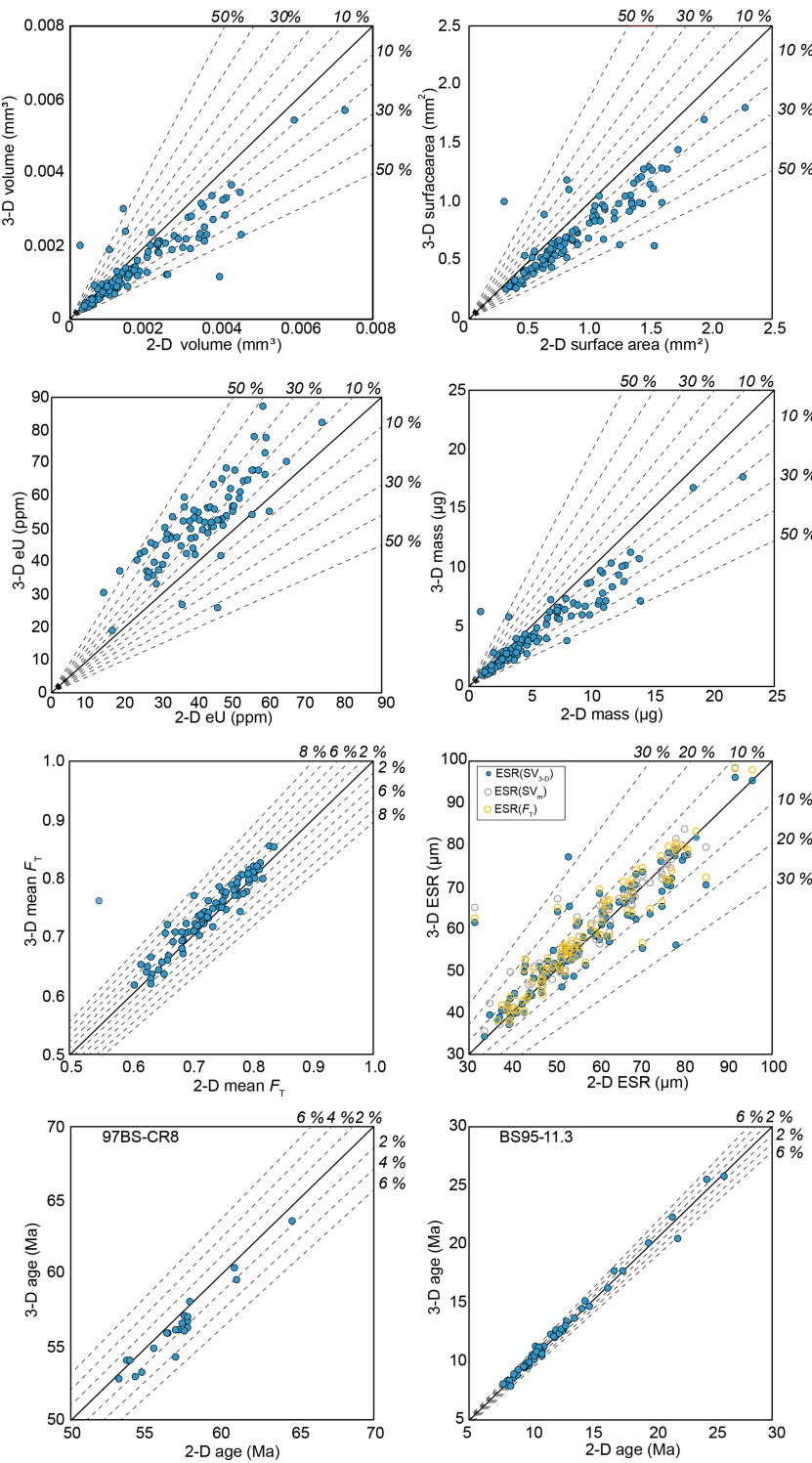

**Figure 5.** Scatter plots for volume, surface area, eU, mass, mean $F_T$, ESR, and age for both samples. Both samples are plotted together unless otherwise noted. Each data point represents a single apatite aliquot. Dashed lines represent the percent difference from the 1 : 1 line (black line). 3-D data measurements are plotted on the $y$ axis in all plots. 2-D measurements overestimate volume, surface area, and mass and underestimate eU and mean $F_T$.

**Table 1.** Comparison of 2-D and 3-D measurement data.

| Sample | $n$ (grains) | Avg. 3-D / 2-D | $1\sigma$ | Absolute difference | | | | |
| | | | | Avg. | $1\sigma$ | Median | Max | Min |
| --- | --- | --- | --- | --- | --- | --- | --- | --- |
| **Length** | | | | | | | | |
| 97BS-CR8 | 49 | 1.00 | 0.03 | 2 | 2 | 2 | 7 | 0.1 |
| BS95-11.3 | 59 | 0.97 | 0.1 | 6 | 7 | 5 | 49 | 0.1 |
| Combined | 108 | 0.98 | 0.1 | 4 | 6 | 3 | 49 | 0.1 |
| **Width** | | | | | | | | |
| 97BS-CR8 | 49 | 1.04 | 0.05 | 15 | 8 | 14 | 32 | 0.2 |
| BS95-11.3 | 59 | 1.03 | 0.09 | 17 | 9 | 17 | 44 | 1 |
| Combined | 108 | 1.03 | 0.07 | 16 | 8 | 16 | 44 | 0.2 |
| **Volume** | | | | | | | | |
| 97BS-CR8 | 49 | 0.85 | 0.25 | 22 | 19 | 18 | 109 | 2 |
| BS95-11.3 | 59 | 0.80 | 0.20 | 24 | 14 | 23 | 80 | 5 |
| Combined | 108 | 0.82 | 0.22 | 23 | 16 | 20 | 109 | 2 |
| **Surface area** | | | | | | | | |
| 97BS-CR8 | 49 | 0.83 | 0.15 | 21 | 10 | 20 | 60 | 4 |
| BS95-11.3 | 59 | 0.80 | 0.12 | 22 | 8 | 21 | 43 | 3 |
| Combined | 108 | 0.81 | 0.14 | 22 | 9 | 20 | 60 | 3 |
| **Mass** | | | | | | | | |
| 97BS-CR8 | 49 | 0.85 | 0.25 | 22 | 19 | 18 | 109 | 2 |
| BS95-11.3 | 59 | 0.80 | 0.20 | 24 | 14 | 23 | 80 | 5 |
| Combined | 108 | 0.82 | 0.22 | 23 | 16 | 20 | 109 | 2 |
| **ESR ($SV_m$)** | | | | | | | | |
| 97BS-CR8 | 49 | 1.03 | 0.04 | 4 | 4 | 2 | 21 | 0.1 |
| BS95-11.3 | 59 | 1.02 | 0.08 | 5 | 6 | 4 | 32 | 0.03 |
| Combined | 108 | 1.02 | 0.06 | 5 | 5 | 3 | 32 | 0.03 |
| **ESR ($SV_{3\text{-}D}$)** | | | | | | | | |
| 97BS-CR8 | 49 | 1.01 | 0.10 | 6 | 8 | 3 | 45 | 0.04 |
| BS95-11.3 | 59 | 1.00 | 0.08 | 6 | 5 | 4 | 26 | 0.02 |
| Combined | 108 | 1.01 | 0.09 | 6 | 7 | 4 | 45 | 0.02 |
| **ESR ($F_T$)** | | | | | | | | |
| 97BS-CR8 | 49 | 1.03 | 0.04 | 4 | 3 | 4 | 12 | 1 |
| BS95-11.3 | 59 | 1.02 | 0.08 | 6 | 6 | 5 | 28 | 0.1 |
| Combined | 108 | 1.02 | 0.07 | 5 | 5 | 4 | 28 | 0.1 |
| **$F_{T,U}$** | | | | | | | | |
| 97BS-CR8 | 49 | 1.02 | 0.02 | 2 | 1 | 2 | 8 | 0.1 |
| BS95-11.3 | 59 | 1.01 | 0.03 | 2 | 2 | 2 | 9 | 0.1 |
| Combined | 108 | 1.01 | 0.02 | 2 | 2 | 2 | 9 | 0.1 |
| **$F_{T,\text{Th}}$** | | | | | | | | |
| 97BS-CR8 | 49 | 1.01 | 0.02 | 2 | 1 | 1 | 7 | 0.01 |
| BS95-11.3 | 59 | 1.00 | 0.03 | 2 | 2 | 2 | 9 | 0.1 |
| Combined | 108 | 1.00 | 0.03 | 2 | 2 | 1 | 9 | 0.01 |

| Sample | $n$ (grains) | Avg. 3-D / 2-D | $1\sigma$ | Absolute difference | | | | |
|---|---|---|---|---|---|---|---|---|
| | | | | Avg. | $1\sigma$ | Median | Max | Min |
| Mean $F_T$ | | | | | | | | |
| 97BS-CR8 | 49 | 1.02 | 0.02 | 2 | 1 | 1 | 5 | 0.2 |
| BS95-11.3 | 59 | 1.01 | 0.03 | 2 | 2 | 2 | 9 | 0.01 |
| Combined | 108 | 1.01 | 0.02 | 2 | 2 | 2 | 9 | 0.01 |
| eU | | | | | | | | |
| 97BS-CR8 | 49 | 1.24 | 0.12 | 24 | 12 | 22 | 49 | 3 |
| BS95-11.3 | 59 | 1.30 | 0.27 | 33 | 23 | 28 | 112 | 6 |
| Combined | 108 | 1.29 | 0.24 | 31 | 20 | 26 | 112 | 3 |
| Age | | | | | | | | |
| 97BS-CR8 | 49 | 0.99 | 0.01 | 2 | 1 | 1 | 5 | 0.2 |
| BS95-11.3 | 59 | 0.99 | 0.03 | 2 | 2 | 2 | 8 | 0.01 |
| Combined | 108 | 0.99 | 0.02 | 2 | 2 | 2 | 8 | 0.01 |

ESR($SV_m$): BoxA and BoxB assuming hexagonal prism shape, ESR($SV_{3\text{-}D}$): Blob3D volume and SA measurements, ESR($F_T$): $F_T$-equivalent sphere.

ample, if a grain has uneven terminations, it is at the analyst's discretion to measure the longest axis or split the difference, whereas the CT analysis always reflects the longest axis. Similarly, CT scanning is also not subject to any user error introduced by measuring the apatite grain not lying on its widest face or at an incorrect magnification. In our dataset, a couple of grains have very large deviations from the CT-derived volume, which may be caused by the microscope magnification setting being slightly off during measuring. Of course, the degree of analyst error is subject to many factors (e.g., experience of the analyst, the age and type of microscope, measuring software, etc.) and must be addressed on a lab-by-lab basis. In this study we found that human error may lead to "outliers" in the results, and therefore it is a factor that we consider.

## 3.2    Volume and surface area

Volumes and surface areas calculated using the 2-D microscope dimensions both average $\sim 20\,\%$ larger than the 3-D calculations (3-D / 2-D$_{\text{VOL}} = 0.82$, 3-D / 2-D$_{\text{SA}} = 0.81$) (Table 1, Fig. 5). Specifically, 2-D volumes and surface areas calculated from length and width data assuming a hexagonal prism shape have an absolute average difference of $23 \pm 32\,\%$ ($2\sigma$) and $22 \pm 18\,\%$ ($2\sigma$), respectively, from 3-D Blob3D-calculated volumes and surface areas.

## 3.3    ESR and mass

The 2-D ESR is calculated using the surface-area-to-volume ratio (SA/V), which is derived assuming a hexagonal prism with the length and width dimensions measured on the microscope (Eq. 2, Fig. 6). The 3-D data had the ESR calculated

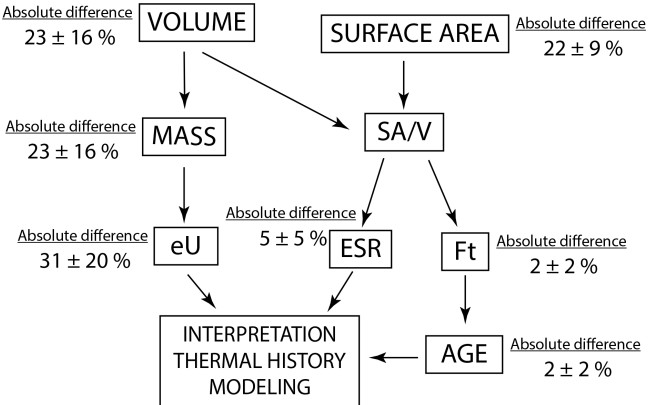

**Figure 6.** Workflow diagram showing the effect of volume and surface area measurements on other parameters used for (U–Th) / He age calculation and interpretation. The average absolute differences between 2-D and 3-D measurements for each of the parameters are reported with their $1\sigma$ uncertainties (reported in Table 1). Note that the greatest deviations are in volume and surface area, as well as parameters that rely on volume alone. ESR, $F_T$, and ages deviate less because they use SA/V, which is $\sim 1$ between 2-D and 3-D measurements.

based on SA/V in three ways. First, the SA/V for ESR$_{SVm}$ is calculated using the BoxA and BoxB values provided by Blob3D and assuming a hexagonal prism, mimicking the 2-D approach. The variation between 2-D and 3-D ESR$_{SVm}$ measurements has a $2\sigma$ spread of $\pm 12\,\%$, but the variability is fairly evenly split in overestimating and underestimating the ESR such that the average 3-D / 2-D ratio is 1.02. Second, the 3-D SA/V is calculated using the surface area and volume measurements output by Blob3D (ESR$_{SV3D}$). The

**Table 2.** (U–Th) / He age results.

| Aliquot | Grain morph. | 3-D $F_T$ age (Ma) | 2-D $F_T$ age (Ma) |
|---|---|---|---|
| Sample 97BS-CR8- | | | |
| *1 | FL PR | 57.2 | 78.8 |
| 2 | PR CH, $i$ | 56.3 | 57.7 |
| 3 | PR PR | 54.1 | 53.8 |
| 4 | PR PR | 56.2 | 57.2 |
| 5 | PR PR | 58.0 | 57.9 |
| 6 | PR PR | 54.1 | 53.9 |
| 7 | PR PR | 60.3 | 60.8 |
| 8 | FL PR | 53.0 | 54.3 |
| 9 | FL PR | 57.1 | 57.5 |
| 10 | PR PR | 56.0 | 56.4 |
| 11 | FL FL, $i$ | 54.9 | 55.5 |
| 12 | FL PR | 63.5 | 64.6 |
| 13 | FL PR | 56.7 | 57.7 |
| 14 | PR PR | 55.9 | 56.4 |
| 15 | PR PR | 56.6 | 57.4 |
| 16 | FL FL | – | – |
| 17 | FL PR | 57.0 | 57.8 |
| 18 | FL CH | – | – |
| 19 | FL PR | 52.8 | 53.2 |
| 20 | PR CH | 49.7 | 51.2 |
| 21 | PR CH | 59.5 | 60.9 |
| 22 | PR PR | 53.3 | 54.7 |
| 23 | PR CH | 56.2 | 57.0 |
| Sample 97BS-CR8- | | | |
| *24 | PR PR | 101.2 | 98.4 |
| 25 | PR PR | 56.1 | 57.5 |
| 26 | PR PR | 54.3 | 57.0 |
| Average | | 56.0 | 56.8 |
| SD | | 2.9 | 2.9 |
| %RSD | | 5.1 | 5.1 |
| Subsample average | | 56.0 | 56.9 |
| Sample BS95-11.3- | | | |
| 1 | PR PR | 10.5 | 10.3 |
| 2 | FL PR | 13.6 | 13.8 |
| 3 | FL PR | 8.6 | 8.8 |
| 4 | PR PR | 13.3 | 13.2 |
| 5 | FL PR | 12.6 | 12.8 |
| 6 | PR PR | 10.8 | 11.1 |
| 7 | PR PR | 10.6 | 10.7 |
| 8 | FL PR | 10.2 | 10.2 |
| 9 | PR PR | 25.5 | 24.7 |
| 10 | PR PR | 12.3 | 11.8 |
| 11 | PR CH | 10.4 | 11.1 |
| 14 | FL PR | 10.8 | 10.8 |
| 15 | FL PR | 9.5 | 10.0 |
| 16 | FL PR | 9.7 | 9.9 |
| 17 | FL CH | 8.7 | 9.1 |
| 18 | PR CH | 9.2 | 9.2 |
| 19 | FL PR | 9.4 | 9.6 |
| 20 | FL FL | 25.7 | 26.1 |

| Aliquot | Grain morph. | 3-D $F_T$ age (Ma) | 2-D $F_T$ age (Ma) |
|---|---|---|---|
| Sample BS95-11.3- | | | |
| 21 | FL PR | 8.8 | 8.8 |
| 22 | PR PR | 8.3 | 8.2 |
| 23 | PR CH, $i$ | 12.6 | 12.2 |
| 24 | FL PR | 10.8 | 10.9 |
| 25 | FL PR | 9.8 | 10.0 |
| 26 | FL FL | 10.1 | 10.5 |
| 27 | FL CH | 15.1 | 14.7 |
| 28 | FL CH, $i$ | 8.2 | 8.4 |
| 29 | PR CH | 10.8 | 10.8 |
| 30 | PR PR, $i$ | 12.5 | 12.8 |
| 31 | PR PR | 12.0 | 12.1 |
| 32 | FL PR | 14.6 | 15.0 |
| 33 | PR PR | 17.6 | 17.1 |
| 34 | PR CH | 11.0 | 10.9 |
| 35 | PR PR | 12.1 | 12.1 |
| 36 | PR CH, $i$ | 9.5 | 9.7 |
| 37 | FL FL | 10.4 | 10.5 |
| 38 | FL CH | 9.6 | 9.8 |
| 39 | PR PR, $i$ | 12.3 | 12.5 |
| 40 | PR PR | 7.9 | 8.3 |
| 41 | PR CH | 22.3 | 21.8 |
| 42 | PR PR | 11.2 | 11.3 |
| 43 | FL CH | 9.9 | 10.0 |
| 44 | PR PR | 9.7 | 9.8 |
| 45 | FL CH | 8.0 | 7.9 |
| 46 | PR PR | 11.0 | 11.0 |
| 48 | PR PR | 20.0 | 19.9 |
| 49 | PR CH, $i$ | 17.6 | 17.8 |
| 50 | PR PR, $i$ | 11.2 | 10.6 |
| 51 | FL PR, $i$ | 14.5 | 14.4 |
| 52 | FL CH | 12.7 | 12.7 |
| 53 | FL PR | 12.9 | 13.0 |
| 54 | PR CH | 9.8 | 10.0 |
| 55 | PR PR | 9.6 | 9.6 |
| 56 | FL CH | 16.1 | 16.6 |
| 57 | PR PR | 11.1 | 10.8 |
| 58 | PR CH, $i$ | 9.5 | 9.6 |
| 59 | PR PR | 20.4 | 22.3 |
| 60 | PR PR | 7.8 | 8.5 |
| 61 | FL CH | 10.8 | 10.7 |
| 62 | CH CH | 10.6 | 11.1 |
| Average | | 12.1 | 12.2 |
| SD | | 4.0 | 4.0 |
| %RSD | | 33.2 | 32.8 |
| Subsample average | | 12.2 | 12.1 |

FL: flat, PR: prismatic, CH: chipped or broken, $i$: inclusion, ∗ excluded from average, SD.

**Table 3.** 2-D vs. 3-D data comparison with other studies.

| This study: 108 grains, CT pixel 4–5 μm | | | | |
| --- | --- | --- | --- | --- |
| | Avg. 3-D / 2-D | $1\sigma$ | % diff. avg. | $1\sigma$ |
| Volume | 0.82 | 0.22 | 23 | 16 |
| Surface area | 0.81 | 0.14 | 22 | 9 |
| ESR | 1.02 | 0.06 | 5 | 5 |
| Mass | 0.82 | 0.22 | 23 | 16 |
| $F_T$ | 1.01 | 0.02 | 2 | 2 |

| Evans et al. (2008): four grains, CT pixel 3.77 μm | | | | |
| --- | --- | --- | --- | --- |
| | Avg. 3-D / 2-D | $1\sigma$ | % diff. avg. | $1\sigma$ |
| Volume | 0.68 | 0.09 | 32 | 9 |
| Surface area | 0.77 | 0.08 | 23 | 8 |
| ESR | – | – | – | – |
| Mass | 0.66 | 0.08 | 34 | 8 |
| $F_T$ | 0.93 | 0.10 | 7 | 10 |

| Glotzbach et al. (2019): 24 grains, CT pixel 1.2 μm | | | | |
| --- | --- | --- | --- | --- |
| | Avg. 3-D / 2-D | $1\sigma$ | % diff. avg. | $1\sigma$ |
| Volume | 1.04 | 0.2 | 15 | 13 |
| Surface area | 1.12 | 0.17 | 16 | 14 |
| ESR | 0.93 | 0.06 | 8 | 5 |
| Mass | – | – | – | – |
| $F_T$ | 0.99 | 0.02 | 2 | 2 |

variation between 2-D and 3-D $ESR_{SV3D}$ is even larger at $\pm 18\%$ ($2\sigma$), with an average 3-D / 2-D ratio of 1.01 (Table 1, Fig. 5).

The $F_T$-based ESR was on average similar to the SV-based one ($ESR_{F_T}$ / $ESR_{SVm}$ = 1.0), but the variation was $\pm 9\%$ for the two samples, and extreme values were 9 % higher and 21 % lower. The relative variation of the $ESR_{F_T}$ value with the 2-D data is $\pm 14\%$, similar to that for the other 3-D ESR calculations (Table 1, Fig. 5).

The grain mass is calculated from the volume data using a nominal apatite density, and therefore 2-D and 3-D mass determination directly reflect the variability in the 2-D and 3-D volume data. The 2-D approach consistently overestimates the mass, with a high degree of scatter (3-D / 2-D = $0.82 \pm 0.44$ ($2\sigma$)) (Table 1, Fig. 5).

### 3.4   $F_T$ corrections

$F_{T,U}$ and $F_{T,Th}$ correction factors calculated from the 2-D data are generally 1 %–2 % lower than the Blob3D U and Th $F_T$ factors. To combine the $F_T$ factors into a single term that is applied to the (U–Th) / He age, a mean $F_T$ was calculated in two ways using Eq. (6) (see Methods). This results in mean $F_T$ factors that vary by an average of 2 % between the 2-D and 3-D datasets. The $1\sigma$ scatter in 3-D / 2-D $F_T$ factors is 1.8 %, though individual differences can reach up to 9 % (Table 1, Fig. 5).

### 3.5   (U–Th) / He age and effective uranium

We calculated the apatite (U–Th) / He-corrected age by dividing the raw (U–Th) / He age by the mean $F_T$ factor. The 2-D $F_T$ (U–Th) / He ages tend to be slightly older than the 3-D $F_T$ (U–Th) / He ages (3-D / 2-D = 0.99) owing to the fact that the 2-D $F_T$ values are slightly lower, leading to a larger correction (Table 1, Fig. 5). The average difference between the 2-D and 3-D $F_T$-corrected ages is 2 %, mimicking that of the variation between 2-D and 3-D $F_T$ (full range is < 1 % to 9 %). This has an insignificant impact on the mean age and uncertainty for both samples. Sample 97BS-CR8 has a 2-D $F_T$ mean age of $56.8 \pm 2.9$ Ma and a 3-D $F_T$ mean age of $56.0 \pm 2.9$ Ma (Table 2, Fig. 5). Sample BS95-11.3 has a 2-D $F_T$ age of $12.2 \pm 4.0$ Ma and a 3-D $F_T$ mean age of $12.1 \pm 4.0$ Ma (Table 2, Fig. 5).

The effective uranium concentrations (eU = [U] + [Th] $\times 0.238+$ [Sm] $\times 0.0012$) for the apatite are normalized to the mass of the grain. Since 2-D and 3-D grain mass calculations varied by $\sim 25\%$, the eU concentration measurements vary by a similar degree (3-D / 2-D = $1.29 \pm 0.24$ ($2\sigma$)) (Table 1, Fig. 5). Note that not all grains were analyzed for U, Th, and Sm, so there are less data for eU comparison than mass.

## 4   Discussion

### 4.1   Accuracy of 2-D vs. 3-D grain measurements

#### 4.1.1   Volume and surface area

One of the main motivations behind this study was to assess the accuracy of 2-D grain measurements and using an assumed grain geometry for calculating grain parameters (volume, ESR, mass, $F_T$) and the impact on the accuracy of the final (U–Th) / He age and eU. For this reason, we selected two samples from crystalline basement rocks that experienced relatively fast exhumation and no significant subsequent reheating in order to reduce the impact of geologic or kinetic factors that could lead to age dispersion.

The most striking deviations between 2-D and 3-D measurements are in the volume and surface area, which 2-D measurements consistently overestimated by 20 %–25 % in our study, with a large degree of scatter ($1\sigma = 22\%$ and 14 %, respectively). These results are in line with previous work. Evans et al. (2008) observed a similar discrepancy in the five apatite grains they measured: their 2-D-based volumes were 30 % greater than the 3-D volumes (Table 3). Our dataset contains > 100 apatite grains, implying that the 2-D overestimation of volume (and therefore mass) may be systematic in the 2-D measurement approach. In contrast, Glotzbach et al. (2019) analyzed 24 apatite grains and found that the 2-D volume measurements varied by a similar magnitude ($\sim 15\%$) but did not systematically overestimate the volume as in our study and Evans et al. (2008) (Table 3). This is likely due in large part to their procedure of measuring

three dimensions and selecting the appropriate shape model on a grain-by-grain basis, including ellipsoids for anhedral grains and accounting for terminations using the functions provided in Ketcham et al. (2011), rather than assuming exclusively flat-terminated hexagonal prisms.

There are multiple factors that can contribute to overestimating the volume of a given apatite crystal. First, the assumption of a hexagonal prism crystal shape with flat terminations, in which the length of the grain is used as the height of the prism, has the potential to overestimate the volume if the crystal has tapered ends (Fig. 4). However, our data suggest this can only account for about a third of the volume difference because even crystals with two flat (or broken) ends still had an average volume difference of 13 %. Second, the ideal prism model also presumes a perfect, equal-sided hexagonal cross section perpendicular to the $c$ axis, for which the ratio of width to height should be $2/\sqrt{3}$, or 1.1547. The 3-D shape measurements give mean ratios of 1.25(02) and 1.23(01) for our two samples, indicating that the cross sections are on average flatter than ideal hexagonal prisms. The nonideality of this cross section was also noted by Glotzbach et al. (2019) and can result in either an underestimate or overestimate of volume, depending on which face the grain is lying on when measured in 2-D. The systematic bias we observe is not surprising as apatites commonly come to rest on their flatter side, whereas some of our observed scatter comes from this not always being the case. We estimate that this shape divergence explains about a quarter of the departure between 2-D and 3-D volume in our data. The remaining deviation may be due to chipped crystals, surface roughness, or other deviations from a perfect prism that the 2-D calculation cannot account for.

A number of factors will directly impact surface area calculations. Surface area is calculated from the 2-D measurements by assuming a perfectly smooth prism. CT has the potential to capture irregular surfaces present in natural apatite grains, which if present and resolution is sufficient, should lead to higher surface area calculations in the 3-D data. However, surface area is problematic to measure in CT data, regardless of resolution. Irregular surfaces are to some degree fractal entities, making their measured areas dependent on measurement scale, and the "correct" answer is not straightforward to define. All CT images are naturally blurry to some extent, smoothing out both irregularities and also sharp corners and edges. Conversely, the 3-D measurement process of segmentation by thresholding can lead to artificial enhancement of surface area due to voxelation effects (the 3-D equivalent of pixilation).

In our data, the 2-D measurements consistently result in a higher surface area than the 3-D measurements. This is probably partly due to the $\sim 5\,\mu m$ resolution of our CT data and also to the flat-terminated hexagonal prism model leading to an overestimate. Evans et al. (2008) observe a similar discrepancy in surface area measurements between 2-D and 3-D data (2-D $\sim 23\,\%$ higher) with a 3.77 $\mu m$ resolution scan

(Table 3). On the other hand, Glotzbach et al. (2019) scanned their grains at a 1.2 $\mu m$ resolution and their 2-D measurements gave surface areas on average 8 % lower than 3-D (Table 3). As with volume, a large part of the difference is probably due to their using a more accurate shape model than an ideal equal-sided hexagonal prism. The overshoot may be in part due to their higher CT data resolution capturing roughness better, but their 3-D images also show voxelation effects such as ridge sets on flat surfaces that likely increased their surface areas to an unknown extent.

We note that the nature of the alpha stopping process, both in reality and as simulated, makes it essentially a $\sim 20\,\mu m$ smoothing filter, so short-length-scale roughness has a negligible effect on alpha particle retention and $F_T$ calculation. This point is demonstrated by our sensitivity analysis (Appendix B), which shows that a bumpy, voxelated sphere has the same $F_T$ correction as a perfect, smooth one. Thus, while surface area is difficult to measure precisely in general, it is unimportant to measure precisely for this application.

### 4.1.2 Mass and eU

The discrepancy in volume between 2-D and 3-D measurements directly impacts the mass calculation, causing the grain masses derived from the 2-D measurements to be $\sim$ 25 % higher than the 3-D grain mass determinations (Fig. 6). Evans and others (2008) found similar deviations, with their masses calculated from 2-D volumes $\sim 30\,\%$ greater than their masses for 3-D volumes (Table 3). Both of these divergences stem from using the assumption of a flat-ended hexagonal prism, whereas an approach that takes grain shape into account when choosing the $F_T$ formula (Ketcham et al., 2011; Glotzbach et al., 2019) avoids this systematic bias. However, in all cases that use perfect shape models, the relative scatter is on the order of 20 % ($1\sigma$), which is high enough to be worth fixing.

Although the age equation does not require knowledge of the grain volume or mass, both are necessary to calculate reported concentrations for U, Th, Sm, and He (Fig. 6). The U, Th, and Sm concentrations, often combined into a single term, "effective uranium" (eU), have been used a proxy for radiation damage within a crystal, and age versus eU correlations are commonly used for interpretation of age scatter and thermal history inverse modeling (e.g., Flowers et al., 2009; Guenthner et al., 2013; Fox et al., 2019). Therefore, accurate knowledge of volume has cascading effects from mass to eU concentration and age interpretation (Fig. 6). Comparison between eU calculated for the 3-D mass data and 2-D mass data shows that the 2-D masses underestimate the bulk eU concentrations by $\sim 20\,\%$–30 %. This is consistent with the 2-D mass data being $\sim 25\,\%$ higher than the 3-D mass data, which would have the effect of "diluting" any eU signal; moreover, the much higher degree of scatter in the mass data caused by 2-D analysis ($\pm 44\,\%$ ($2\sigma$)) can be expected to muddy any age–eU correlation that may be present.

### 4.1.3 ESR

The various ESR calculations all yielded similar results on average but high degrees of variation between measurement and calculation modes (5 %–6 %). In addition to being more accurate for simplifying complex shapes to spheres for diffusion calculations, the $\mathrm{ESR}_{F_T}$ method is also likely more robust than others that presume or measure surface area. Surface area, beyond being difficult to define and measure for irregular natural objects in a resolution-resistant way, has only secondary importance for diffusion and $F_T$ calculations when it varies on a fine scale compared to the grain (i.e., micrometer-scale roughness). Analogously with mass, excess variation in ESR ($\pm 14$ % ($2\sigma$)) can degrade age–size correlations.

### 4.1.4 $F_T$

A somewhat surprising result of our study is that, despite volume and surface areas being very different between the 2-D and 3-D methods, these differences largely canceled each other out in S/V-based $F_T$ calculations. This is in large part because volume and surface area covary, both in the assumed models and the actual measurements, so an error in one leads to a similar magnitude of error in the other (Fig. 6).

A result that more closely conformed to expectation is that, as grain size fell, dispersion between 2-D and 3-D $F_T$ values increased, although it remained modest. The standard deviation of 3-D / 2-D $F_{T,U}$ was 2.7 % for grains with $F_{T,U}$ values from 0.6 to 0.7, 2.4 % from 0.7 to 0.8, and 1.3 % for grains above 0.8.

While the above comparison takes into account 24 to 53 grains per sample, most applications of (U–Th) / He analyze 3–5 grains per sample. As a more practical comparison of the difference between 2-D and 3-D Mean $F_T$, we randomly subsampled the average of four grains from our results 1000 times (Fig. 7). We found that even when subsampling four grains, $\sim 90$ % of runs had a mean deviation in 3-D / 2-D $F_T$ less than 3 %.

## 4.2 Reproducibility of (U–Th) / He ages

In addition to assessing the accuracy of using the 2-D measurements, this study aimed to quantify the uncertainties that may be introduced by such measurements, particularly in $F_T$, as a means to potentially improve age accuracy, precision, and intra-sample dispersion. Previous studies have estimated that uncertainties in $F_T$ calculation can account for 1 %–5 % of sample age uncertainty (Evans et al., 2008; Glotzbach et al., 2019). Our results are consistent with this range and suggest that uncertainties in the U and Th $F_T$ calculation are on the order of 1 %–3 %, and mean $F_T$ varies by 2 % (Table 1). We find the greatest deviations are likely caused by user error for our samples and not the assumed grain geometry. In samples with less euhedral apatite grains, the effects of $F_T$ and an assumed grain geometry can increase.

Our data also show that the 3-D $F_T$ correction does not increase the overall sample age precision for the samples in this study. For sample 97BS-CR8, 24 apatite grains were analyzed, two of which are outliers. Of the two outliers, one (97BS-CR8-1) was clearly caused by a user error during microscope measurement, leading to an incorrect $F_T$ correction (0.55) and old age (78.8 Ma). This was discovered during 3-D image processing, in which the same grain was identified, measured correctly, and produced an $F_T$ of 0.76 and a more congruent corrected age of 57.2 Ma. In contrast, for a second outlier (97BS-CR8-24), the 2-D and 3-D $F_T$-corrected ages both produced anomalous ages of 101.2 and 98.4 Ma, respectively. An unusually high He concentration is the likely culprit for the old age for this grain, but its cause is not evident from our data. Excluding these two outliers, the average age and uncertainty for the sample population ($n = 22$ grains) calculated based on the 2-D and 3-D measurements are indistinguishable ($56.8 \pm 2.9$ and $56.0 \pm 2.9$ Ma); relative errors are 5.1 % in both cases.

Similarly, the sample ages calculated with 3-D and 2-D data for 95BS-11.3 (n=59 aliquots) are indistinguishable at $12.2 \pm 4.0$ and $12.1 \pm 4.0$ Ma, respectively. Unlike sample 97BS-CR8, there was no clear-cut evidence of user error, and the relatively high age uncertainty (33 %) is reproducible between the 2-D and 3-D $F_T$-corrected ages. Five aliquots produced ages > 20 Ma, which skews the mean age older (the median age is 10.2 Ma, within the error of the previous reported age in Stockli et al., 2002). The apatite ages do not correlate with factors such as ESR (grain size) or eU. The > 20 Ma aliquots all have high He concentrations (nmol g$^{-1}$) compared with the bulk of the sample, suggesting excess He, possibly due to implantation from high U–Th neighbors, or the presence of undetected and insoluble high eU inclusions.

In addition to the above calculations, we randomly subsampled four grains 1000 times to assess the variability in $F_T$-corrected age for a number of grains that is more comparable to other studies. The results are plotted in Fig. 7 and reported in Table 2. The mean of the 1000 trials is indistinguishable from the entire analyzed population.

Overall, these data suggest that although the 3-D $F_T$ can provide a more accurate $F_T$ correction and varies from 2-D estimations by $\sim 2$ %, it has a minimal effect on the calculated sample age (1 %–2 %) and no effect on the reproducibility for these two samples. This is not surprising, as a $\sim 2$ % error would constitute a negligible proportion of the often-cited 6 % dispersion derived from analyzing age standards; error propagation indicates that removing a source of 2 % error would only reduce an overall 6 % error to 5.7 %. This points to the importance of other factors in intra-sample dispersion, such as U–Th zonation, and/or excess He from nano-inclusions or high U–Th neighbors.

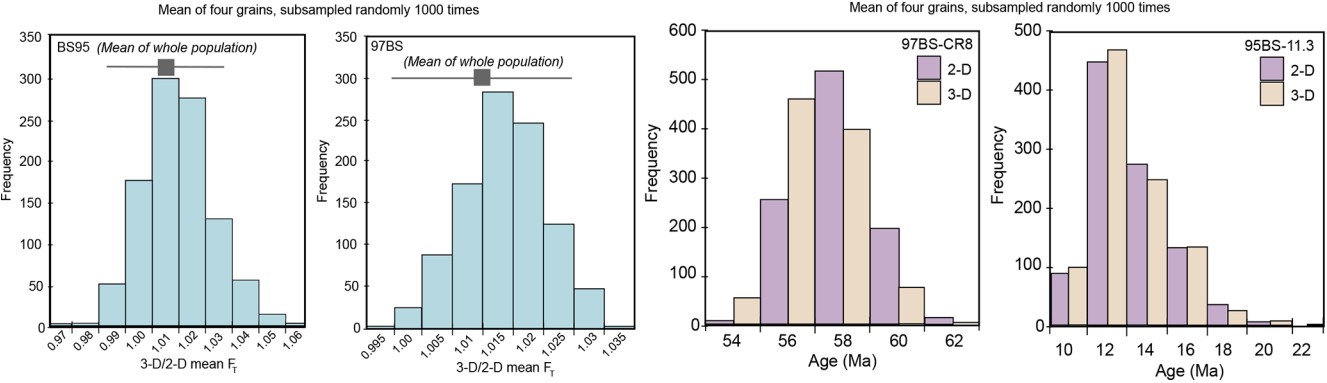

**Figure 7.** Histograms showing the 3-D / 2-D mean $F_T$ and 2-D and 3-D (U–Th) / He age. Histograms show the results of randomly subsampling four grains from each sample 1000 times. For mean $F_T$ and sample age, the subsampling is indistinguishable from the mean of the whole population analyzed. Note: numbers on $x$ axis refer to the bars (bins) and not the tick marks.

## 4.3 Effects of inclusions or broken grains

It is widely accepted that inclusions and broken grains are both contributors to intra-sample dispersion and inaccurate He ages, particularly anomalously old ages. Inclusions in apatite can act as He traps or a source for excess He, particularly mineral inclusions that do not dissolve during apatite $HNO_3$ digestion (e.g., Ehlers and Farley, 2003). Both apatite samples had multiple grains with high-density and low-density inclusions detectable by microscope during picking and/or the CT scan (Fig. 2). In both samples, the presence of inclusions did not have any discernable effect on the (U–Th) / He age (Table 2). While inclusions are certainly a source for error and dispersion in many samples and should be avoided, at least the easily visible ones do not appear to be relevant in these samples, which suggests they are likely also not U–Th-bearing inclusions. For future studies, an added benefit of CT is the detection of high- and low-density mineral and fluid inclusions.

Similarly, broken grains can be a source of dispersion if they were broken after the sample passed through the He partial retention zone, e.g., after the grain began to accumulate He (see Beucher et al., 2013; Brown et al., 2013). Typically, this may occur during erosional transport or during mineral separations. Brown et al. (2013) estimate that broken grains can contribute 7 to > 50 % dispersion from the sample age, depending on cooling history. In our samples, grain terminations varied from doubly prismatic to flat and in some cases appeared chipped or broken. However, there is no clear correlation between the chipped or broken grains and He age (see Table 2). One possibility is that the grains broke prior to cooling through the He retention zone. This seems somewhat unlikely, given that both samples come from crystalline rocks. Alternatively, and perhaps more plausibly, the variety of crystal habits may reflect how the crystals grew in the host rock. In any case, the grains in these samples that appear to

be chipped or broken are not obvious sources for the age dispersion observed in the samples.

## 4.4 Benefits and limitations of X-ray CT over microscope measurements

This study purposefully selected "high-quality" apatite from fast-cooled plutonic samples to quantify the base uncertainty introduced by 2-D measurements and grain shape assumptions on $F_T$ and (U–Th) / He age factors. Although we found that 3-D grain characterization techniques did not reduce intra-sample age dispersion in our samples, it is still highly probable that the 3-D approach can improve dispersion in samples with less euhedral apatite and more complicated geologic histories. Furthermore, CT scanning mineral grains for (U–Th) / He chronometry has both analytical and practical benefits that go beyond grain measurement. CT provides more accurate grain volume measurements, which becomes increasingly important as grain shapes deviate from idealized forms (e.g., abraded or broken grains). CT data are able to highlight inclusions or other internal heterogeneities based on contrasts in density in the X-ray data, which may not be visible by the naked eye. Furthermore, the CT-mounting method and scanning conditions outlined in this study allow for the scanning of up to 250 grains in a single session, and potentially many more, making it cost and time effective. Different mineral phases can be scanned together, and data can be processed in a batch so that from a single scan, one can gather volume, surface area, caliper dimensions, $F_T$, mass, and ESR at once for several samples and phases. Furthermore, the 3-D $F_T$ and $F_T$-based ESR capabilities of the Blob3D software introduced in this study make batch processing the CT data straightforward. Thus, an analyst will be able to image, characterize, and quantify hundreds of mineral grains in significantly less time than conventional microscope measuring. We anticipate that more volume-based shape measurements can and will be devel-

oped to automatically and quantitatively evaluate grains for euhedrality, rounding, broken faces, and a wealth of other potentially informative data.

CT scanning mineral grains used for (U–Th) / He dating also has the benefit of removing many possible sources of user error during the grain measurement step. Unlike with microscope measurements, the orientation of the apatite grain on the CT mount does not matter, and there is no need to set a magnification or trace the dimensions of the grain by hand, reducing the potential for mistakes. CT also eliminates variability that may arise from different microscopes, lighting conditions, and imaging software, and it creates a digital archive of 3-D grain shapes, densities, and internal structures that a microscope photo cannot capture.

The one required user input to our method is specifying the threshold CT number for grain measurement, for which we recommend using the midpoint value between the mineral and the surrounding medium (e.g., air, epoxy). When scan resolution is low in terms of both voxel size and sharpness, additional care is required; if edge blurring approaches the center of a grain, an alternative thresholding or segmentation procedure may be necessary to obtain accurate volumes (Ketcham and Mote, 2019). We thus do not recommend pushing resolution limits too far; voxel sizes generally should not exceed 1/8 to 1/10 of the shortest dimension of a grain. CT measurement accuracy also requires that the scans be as free as possible from artifacts that cause local changes in CT numbers, such as beam hardening, photon starvation, or rings. We further note that software artifact corrections can sometimes introduce secondary artifacts that may be harder to recognize but still affect calculations (Ketcham and Carlson, 2001), so care is required in the scanning process.

The main limitation of using CT is access to the instrumentation and cost for sample analysis. However, CT scanners are becoming more common as desktop instruments in earth science departments, and many universities have imaging facilities that include micro-CT. As CT instruments continue to proliferate and costs continue to fall, we anticipate that measuring, screening, and documenting grains used for thermogeochronology will become a widely used practice.

## 5   Conclusions

The shape and size of 109 apatite grains from two rapidly cooled plutonic samples were analyzed by 2-D and 3-D methods. 2-D length and width measurements made on an optical microscope were used to calculate surface area, volume, ESR, mass, and $F_T$ assuming an ideal equal-sided, flat-terminated hexagonal prism grain shape. The same apatite crystals were scanned using X-ray computed tomography at a 4–5 μm resolution, and the same factors were calculated using Blob3D software, which does not require assuming a grain shape. A total of 83 new apatite (U–Th) / He ages were collected to resolve the influence of 2-D versus 3-D $F_T$ cor-

rection factors on final (U–Th) / He age and reproducibility. With these data, we derive the following conclusions.

1. Deviations between 2-D and 3-D measurements were greatest in volume and surface area ($\sim 25\%$), which caused mass and eU calculations to deviate by a similar magnitude. Volume and surface area measurements also showed high dispersion of 44 % and 28 % ($2\sigma$), respectively. These sources of scatter weaken the ability to use age–eU and age–size correlations to help interpret age distributions.

2. 2-D $F_T$ measurements only contribute $\sim 2\%$ error on average, even with the erroneous assumption of an ideal grain shape.

3. Inclusions and broken or chipped ends did not have a discernible impact on the (U–Th) / He age dispersion in these samples.

4. The combined (U–Th) / He ages for each sample were indistinguishable for 2-D and 3-D $F_T$ corrections. Similarly, the amount of intra-sample dispersion was identical (both $> 5\%$). This implies that factors other than $F_T$ dominate the intra-sample age uncertainty.

In addition, we present a bulk scanning method that easily allows for the analysis of $> 250$ grains in a single session, new Blob3D software 3-D $F_T$ and shape measurement functions, and new calculations for eU and $ESR_{F_T}$.

**Code and data availability.** The code and data are available in the Supplement to this paper. CT data are archived at https://www.digitalrocksportal.org/projects/216 and https://doi.org/10.17612/CZYH-KC13 (Ketcham and Cooperdock, 2019).

## Appendix A: Calculating $ESR_{F_T}$, mean $F_T$, and eU

### A1 $ESR_{F_T}$ and mean $F_T$

The starting point for calculating the equivalent $F_T$ sphere radius ($ESR_{F_T}$) when $F_T$ values are provided for each decay chain is the $F_T$ equation for a sphere (Farley et al., 1996; Ketcham et al., 2011):

$$F_T = 1 - \frac{3}{4}\frac{S}{R} + \frac{B}{16}\left(\frac{S}{R}\right)^3, \tag{A1}$$

where $R$ is the sphere radius, $S$ is stopping distance, and $B$ is an adjustment factor for the 3rd-degree polynomial term to account for $S$ being the weighted mean of stopping distances along branching decay chains rather than a single stopping distance. For U and Th decay chains $B$ should be 1.31, and for single stopping distances it should be 1 (Ketcham et al., 2011).

Solving this equation for $S/R$ over the $F_T$ range from 0.5 to 1 using a 3rd-degree polynomial to match the effect of the cubic term gives

$$\frac{S}{R} = 1.681 - 2.428F_T + 1.153F_T^2 - 0.406F_T^3 \, (B = 1.31), \tag{A2a}$$

$$\frac{S}{R} = 1.580 - 2.102F_T + 0.801F_T^2 - 0.279F_T^3 \, (B = 1). \tag{A2b}$$

The polynomial in Eq. (A2a) is the appropriate one to use for data to be reported in age tables; Eq. (A2b) is provided for completeness and may be useful for comparing to other calculations that use mean $S$ values to represent chains.

The $F_T$ value to use is the weighted mean incorporating the separate factors $F_{T,238}$, $F_{T,235}$, and $F_{T,232}$, accounting for different alpha productivity along each chain. Expanding the approach of Farley (2002) to account precisely for $^{235}U$, we calculate

$$A_{238} = (1.04 + 0.247\left[Th/U\right])^{-1}, \tag{A3a}$$

$$A_{232} = (1 + 4.21/\left[Th/U\right])^{-1}, \tag{A3b}$$

so that the weighted mean, $\overline{F_T}$, is

$$\overline{F_T} = A_{238}F_{T,238} + A_{232}F_{T,232} + (1 - A_{238} - A_{232})$$
$$F_{T,235}. \tag{A4}$$

Solving the result of Eq. (A2) for $ESR_{F_T}$ requires the analogous calculation to determine the weighted mean stopping distance, $\overline{S}$:

$$\overline{S} = A_{238}S_{238} + A_{232}S_{232} + (1 - A_{238} - A_{232})S_{235}, \tag{A5}$$

where $S_{238}$, $S_{235}$, and $S_{232}$ are the weighted mean stopping distances for each decay chain (18.81, 21.80, and 22.25 µm, respectively, for apatite, but the calculation applies to any mineral). Then, combining Eqs. (A2) and (A5) gives

$$ESR_{F_T} = \overline{S}/\left(\frac{S}{R}\right). \tag{A6}$$

### A2 eU

The earliest mention of eU, or effective uranium with respect to He production, we are aware of is in Shuster et al. (2006), who put forward the formula

$$eU = [U] + 0.235[Th], \tag{A7}$$

where brackets indicate composition in parts per million without a detailed description of its derivation. Converting from elemental or isotopic compositions in parts per million to an equivalent alpha particle production rate requires accounting for decay constants, isotopic proportions, alpha particle production, and atomic mass. We calculate the present-day alpha production rate $R_\alpha$ (here: $\alpha$ g$^{-1}$ yr$^{-1}$) as

$$R_\alpha = A\lambda p N/m_a, \tag{A8}$$

where $A$ is Avogadro's number, $\lambda$ is the decay constant, $p$ is isotopic proportion, $N$ is the number of alpha particles produced in the decay chain, and $m_a$ is atomic mass. The eU factor is then calculated by dividing the Th and Sm $R_\alpha$ by the combined U $R_\alpha$ utilizing the values in Table A1; we find the eU equation to be slightly different:

$$eU = [U] + 0.238[Th] + 0.0012[Sm](\text{or } 0.0083[^{147}Sm]). \tag{A9}$$

We do not know the reason for the small discrepancy with Eq. (A7), but the $\sim 1\%$ difference in the effect of Th is not likely to be important for current uses of eU. The 0.238 factor has a likely uncertainty of $\pm 0.002$; the $^{232}Th$ half-life currently recommended by the nuclear chemistry community has only three significant figures based on a weighted average of several determinations using different methodologies (Browne, 2006; Holden, 1990), whereas the geological community has adopted the value from the single study with the highest reported precision (Le Roux and Glendenin, 1963; Steiger and Jäger, 1977).

We include Sm for completeness, but as its alpha decay has a relatively low recoil energy it is not clear whether simply counting the particle is the most appropriate way to include its potential contribution to damage that affects helium diffusivity. An alternative formulation can be posed in terms of energy deposition (kerma; Shuster and Farley, 2009):

$$R_k = A\lambda p N E/m_a, \tag{A10}$$

where $E$ is the mean alpha particle recoil energy for the decay chain. The revised kerma-based quantity, $eU_k$, is then

$$eU_k = [U] + 0.264[Th] + 0.0005[Sm]$$
$$(\text{or } 0.0034[^{147}Sm]). \tag{A11}$$

This relation predicts that Sm will have an even lower relative contribution to diffusivity than indicated in Eq. (A9), but Th will be 11 % more potent due to its higher mean recoil energy compared to $^{238}U$. We do not currently recommend this approach, but it does pose a potentially testable hypothesis.

**Table A1.** Values used for calculating eU.

|  | $^{238}$U | $^{235}$U | Th | Sm (total) | $^{147}$Sm |
|---|---|---|---|---|---|
| $\lambda^*$ (1 yr$^{-1}$) | 1.55125E−10 | 9.8485E−10 | 4.9475E−11 | 6.539E−12 | 6.539E−12 |
| $P$ | 0.9928 | 0.0072 | 1 | 0.1499 | 1 |
| $m_a$ (g mol$^{-1}$) | 238.029 | 238.029 | 232.039 | 150.36 | 147 |
| $N$ ($\alpha$ and/or chain) | 8 | 7 | 6 | 1 | 1 |
| $\alpha$ g$^{-1}$ yr$^{-1}$ | 3.117E+12 | 1.256E+11 | 7.70E+11 | 3.93E+09 | 2.68E+10 |
| eU factor |  |  | 0.238 | 0.0012 | 0.0083 |
| $E$ (MeV) | 5.359 | 5.946 | 5.990 | 2.247 | 2.247 |
| Energy deposited (g$^{-1}$ yr$^{-1}$) | 1.671E+13 | 7.468E+11 | 4.61E+12 | 8.82E+09 | 6.02E+10 |
| eU$_k$ factor |  |  | 0.264 | 0.0005 | 0.0034 |

* Values for U and Th from Steiger and Jäger (1977).

## Appendix B: Evaluation of accuracy and precision in Blob3D $F_T$ calculations

This Appendix describes a series of tests that demonstrate the accuracy and precision of the methods for $F_T$ calculations implemented in Blob3D (Ketcham, 2005). All calculations are performed in Blob3D or with scripts in IDL, the computer language in which Blob3D is written.

### B1 Centered spheres

In the first set of tests, we use spheres, which Herman et al. (2007) recognized as a good test shape because its surface is poorly approximated by coarse stacked cubes. We begin with a $128^3$-voxel field, and select all voxels with centers within 63 voxel widths of the center of the volume, creating a 63 µm radius sphere with a 1-voxel-thick black boundary on all sides. Four additional lower-resolution versions were then created by rebinning the original dataset to make volumes with $64^3$, $32^3$, $16^3$, and $8^3$ voxels; these datasets were then padded with an additional layer of black (nonselected) voxels on three sides to ensure the spheres had a black boundary on all sides for Blob3D processing. In the 8-bit data volumes, selected voxels have a value of 255 (white) and nonselected ones a value of 0 (black).

If the voxel width is 1 µm in the $128^3$ dataset, the resulting ideal sphere radius is 63 µm, which has an $F_{T,238}$ correction of 0.7777 (stopping distance 18.81 µm). Because of voxelation effects, the actual volume selected will be slightly different than the ideal case; for example, the volume in the $128^3$ dataset corresponds to an equivalent sphere radius (ESR) of 63.02 µm. With each rebinning step, doubling the voxel size roughly maintains the original volume, simulating lower resolution; i.e., 2 µm voxels for the $65^3$-voxel dataset, 4 µm for $33^3$, 8 µm for $17^3$, and 16 µm for $9^3$. We ran an initial set of tests using these voxel sizes and an additional set with the voxel size halved, corresponding to a 31.5 µm radius crystal, close to the lower end of the practical limit ($F_{T,238} = 0.5655$).

Because the calculation employs a Monte Carlo algorithm, answers change slightly from run to run, so for each dataset and resolution results from five Blob3D runs were used to gauge precision. Results are provided in Table B1 and shown in Fig. B1 as the mean measured (calculated) $F_T$ divided by the ideal value for the ESR of the volume actually selected at each resolution, with bars showing 1 standard error.

Results for the 63 µm sphere test are in Table B1a and Fig. B1a. Solid symbols show the result of the normal Monte Carlo analysis, with results accurate to within 0.1 % at up to a 4 µm voxel size, but mean errors rise to approach 1 % with 8 µm voxels. Half-tone symbols show the result of altering the processing by first super-sampling the volume, subdividing each voxel into a $3^3$ set, and then smoothing the expanded data volume with a 5-voxel-wide filter, followed by re-binarizing the data with a threshold (value 127) prior

to the Monte Carlo analysis. This step improves accuracy at 8 µm resolution to within 0.4 % on average and also further reduces the sub-0.1 % error at the 4 µm level. However, the 127 re-threshold value is not the optimal one, as it slightly shrinks the volume due to the overall convex shape of the grain, so the algorithm finds the optimal threshold that reproduces as closely as possible the pre-super-sampled grain volume. The result improves the 8 µm calculation yet more, reducing the mean error to just over 0.2 %, and even with 16 µm voxels the error is only just over 0.5 %. This improvement also demonstrates that getting the volume correct is a primary control on the accuracy of the $F_T$ calculation; this principle is used to examine the case of non-centered spheres later in this Appendix.

Remaining tests use the convention that when voxel sizes are 4 µm or higher the constant-volume super-sampled approach is used; the only cost of super-sampling is slightly more computing time, which is still less than 1 s per grain (but could rise above this level if employed with smaller voxels and larger grains). The 31.5 µm sphere test (Table B1b, Fig. B1b) shows similar results as the larger case; mean errors are less than 0.5 % up to voxel sizes of 8 µm.

### B2 Cylinders

As most apatite (and zircon) grains are elongate, we also tested cylinders as a close-to-worst-case end-member, again because a round outline is more poorly approximated by cubes than a hexagonal or tetragonal one. We created the cylinders by stacking 510 63-voxel-radius circles with blank slices at each end to achieve an aspect ratio close to 4 and down-sampled as with the sphere test four times by powers of 2. Results are shown for the 63 and 31.5 µm cases, with respective ideal $F_{T,238}$ values of 0.8350 and 0.6772, in Table B1c–d and Fig. B1c–d. Even in the coarsest-resolution cases, the mean calculated $F_{T,238}$ values are only off the ideal by 0.3 %.

### B3 Non-centered spheres

In their Monte Carlo $F_T$ implementation, Herman et al. (2007) report poor precision for small spheres when their centers are not centered in a voxel, with errors rising to several percent for a 40 µm radius sphere with 6.3 µm voxels across a range of center locations (calculated $F_T$ range $\sim 0.58$–0.67). Errors of this magnitude correspond to the effect of getting the radius wrong by plus or minus almost an entire voxel.

We tested for voxelation effects on dimensional measurements by running 100 000 trials randomizing the location of the sphere center in a voxel grid using the same radius and voxel size, once again selecting all voxels with centers within the radius of the randomized center. Converting the resulting volumes to sphere-equivalent radii, we got a mean radius error of 0 %, maximum radius errors of $+0.8$ and $-1.1$ %, and

a standard deviation of 0.2 %. At 40 µm (a severe case) a 1 % change in radius leads to a ±0.5 % change in $F_{T,238}$ (range 0.6494–0.6561). Together, these results indicate that the degree to which a sphere is off-center to the CT voxel grid has only a very small effect on its measured size and a correspondingly smaller effect on the $F_T$ determination.

There is a case in which resolution is a concern, however, which is when the grain size approaches the "true" data resolution. All CT data are blurry to some extent due to the finite size of the X-ray focal spot and detector elements, among other factors (ASTM, 2011). This blurring can be characterized as a point-spread function (PSF), which can be considered as a smoothing kernel that "blurs" reality as the CT process translates it into a voxel grid. If the smoothing function width, which can be roughly estimated as the number of voxels it takes to fully transition from one material into another across a flat interface (Ketcham et al., 2010), approaches the grain radius, it can affect grain size and shape measurement (Ketcham and Mote, 2019). Typical PSF widths are on the order of 3–5 voxels in most CT data, so as a rule of thumb the voxel size should be limited to less than 20 % of the grain's shortest dimension. Even in this case accurate grain measurements are possible but require additional steps and calibrations, as described by Ketcham and Mote (2019).

We are thus confident that our implementation provides a high degree of accuracy and precision on even very small grains at low resolutions at which voxel sizes are up to 20 % of the radius.

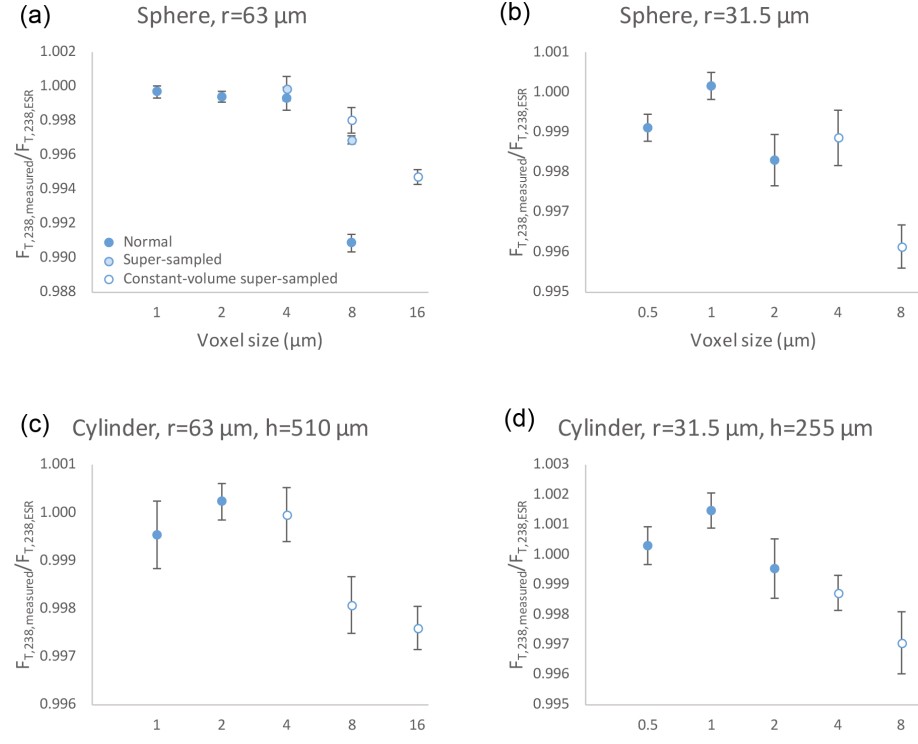

**Figure B1.** Results of Blob3D measurement of synthetic spheres and cylinders.

**Table B1.** Results of Blob3D measurement of synthetic spheres and cylinders.

| Resolution (μm) | [1]Sampling | [2]$ESR_m$ (μm) | [3]$F_{T,238,\text{ideal}}$ | [4]$F_{T,238}$ | $F_{T,238}$ | $F_T/F_{T,\text{ideal}}$ |
|---|---|---|---|---|---|---|
| Sphere radius = 63 μm | | | | | | |
| 1 | normal | 63.02 | 0.7778 | 0.7776(03) | 0.0006 | 0.9997(03) |
| 2 | normal | 63.02 | 0.7778 | 0.7773(03) | 0.0006 | 0.9994(03) |
| 4 | normal | 63.06 | 0.7779 | 0.7774(05) | 0.0011 | 0.9993(06) |
| 4 | super | 63.06 | 0.7779 | 0.7779(05) | 0.0012 | 0.9999(07) |
| 8 | normal | 63.03 | 0.7779 | 0.7707(04) | 0.0009 | 0.9909(05) |
| 8 | super | 63.03 | 0.7779 | 0.7754(02) | 0.0004 | 0.9969(03) |
| 8 | super, cv | 63.03 | 0.7779 | 0.7763(06) | 0.0013 | 0.9980(08) |
| 16 | super, cv | 62.61 | 0.7764 | 0.7723(03) | 0.0007 | 0.9947(04) |
| Sphere radius = 31.5 μm | | | | | | |
| 0.5 | normal | 31.51 | 0.5656 | 0.5651(01) | 0.0003 | 0.9991(02) |
| 1 | normal | 31.51 | 0.5656 | 0.5657(05) | 0.0011 | 1.0002(09) |
| 2 | normal | 31.53 | 0.5658 | 0.5649(09) | 0.0019 | 0.9983(15) |
| 4 | super, cv | 31.52 | 0.5657 | 0.5650(04) | 0.0008 | 0.9989(06) |
| 8 | super, cv | 31.31 | 0.5629 | 0.5607(05) | 0.0011 | 0.9961(09) |
| Cylinder; radius = 63 μm, height = 510 μm | | | | | | |
| 1 | normal | – | 0.8350 | 0.8346(06) | 0.0003 | 0.9995(07) |
| 2 | normal | - | 0.8350 | 0.8352(03) | 0.0007 | 1.0002(04) |
| 4 | super, cv | – | 0.8350 | 0.8350(05) | 0.0010 | 1.0000(06) |
| 8 | super, cv | – | 0.8334 | 0.8318(05) | 0.0011 | 0.9981(06) |
| 16 | super, cv | – | 0.8287 | 0.8267(04) | 0.0008 | 0.9976(04) |
| Cylinder; radius = 31.5 μm, height = 255 μm | | | | | | |
| 0.5 | normal | – | 0.6772 | 0.6774(04) | 0.0009 | 1.0003(06) |
| 1 | normal | – | 0.6772 | 0.6781(04) | 0.0009 | 1.0014(06) |
| 2 | normal | – | 0.6770 | 0.6767(07) | 0.0015 | 0.9995(10) |
| 4 | super, cv | – | 0.6740 | 0.6732(04) | 0.0009 | 0.9987(06) |
| 8 | super, cv | – | 0.6651 | 0.6632(07) | 0.0016 | 0.9971(10) |

[1] Sampling is either normal, super-sampled, or super-sampled maintaining constant volume (cv). [2] $ESR_m$: measured equivalent sphere radius, as the voxelated spheres had slightly different volumes than ideal ones. [3] $F_{T,238,\text{ideal}}$: $F_{T,238}$ value (for the [238]U stopping distance for apatite) for the given shape with the voxelated volume and, for cylinders, aspect ratio. [4] $F_{T,238}$: mean measured $F_{T,238}$ value over five trials, with estimated precision in parentheses.

```
; TestSphereVolumes
;
; Generates a series of voxelated spheres with centers at random points within the
; central voxel, and evaluates the effect on apparent sphere size.
;
; For each test, a voxel field is generated, and a random 3D coordinate is generated
; within the central voxel.  The routine then determines all voxels whose centers
; are within the sphere radius of the central coordinate, and reports the resulting
; volume and sphere-equivalent radius.
;
; INPUT PARAMETERS:
;   None (only through Keyword Parameters)
;
; OUTPUT:
;   Prints to console a tab-delimited table.
;   Columns correspond to:
;     Volume: Volume of voxels in selected region
;     Volume/TrueVol: Denominator is true volume for sphere with given radius
;     Abs(1-Vol/TrueVol): Normalization to show fractional error
;     Radius: Equivalent spherical radius of selected region
;     Radius/TrueRad: Denominator is true radius, as entered by user
;     Abs(1-Rad/TrueRad): Normalization to show fractional error
;   Rows correspond to mean, standard deviation, maximum, and minimum of each
;
; KEYWORD PARAMETERS:
;   NUMTESTS: Number of random spheres to generate; default = 100
;   VOXELSIZE: Voxel edge length; units arbitrary, but same as radius; default = 6.3
;   SPHERERAD: Sphere radius; units arbitrary, but same as voxel size; default = 40.
;   SHOW: Set to show an animation of central slice through each voxelated sphere.
;
; CALLING SEQUENCE:
;   TestSphereVolumes, /SHOW, NUMTESTS=100000L, VOXELSIZE=6.3, SPHERERAD=40.
;
; MODIFICATION HISTORY:
;   Written by:  Rich Ketcham, 6 February 2019

Pro TestSphereVolumes, SHOW=show, NUMTESTS=numTests, VOXELSIZE=voxelSize, $
   SPHERERAD=sphereRad, PSF_RAD=psfRad, THRESH_VAL=threshVal

 if NOT Keyword_Set(numTests) then numTests = 100

 headings = ["Volume","Vol/TrueVol","Abs(1-Vol/TrueVol)","Radius", $
  "Rad/TrueRad","Abs(1-Rad/TrueRad)"]
 results = FltArr(N_Elements(headings),numTests)

 if NOT Keyword_Set(voxelSize) then voxelSize = 6.3 ; microns
 if NOT Keyword_Set(sphereRad) then sphereRad = 40. ; microns
 trueVol = (4./3.)*!PI*sphereRad^3
 voxelVol = voxelSize^3
 arrayDim = Ceil(sphereRad/voxelSize)*2+1
 dv = RandomU(seed,3,numTests)-0.5          ; Displacements inside central voxel

 for testNum=0,numTests-1 do begin
; Calculate distance from every voxel to sphere center
   mid = dv[*,testNum] + arrayDim/2.0
   sph = FltArr(arrayDim, arrayDim, arrayDim)
   for i=0,arrayDim-1 do $
    for j=0,arrayDim-1 do $
     for k=0,arrayDim-1 do begin
       sph[i,j,k] = (i-mid[0])^2 + (j-mid[1])^2 + (k-mid[2])^2
     endfor
   sph = sqrt(sph)
; Accept all voxels with centers closer to midpoint than radius (in voxels)
   sph = (sph LE sphereRad/voxelSize)

   if Keyword_Set(show) then begin
    tvscl, Congrid(sph[*,*,arrayDim/2],512,512)
    Wait, 0.1
   endif

   vol = Total(sph)*voxelVol
   esRad = (vol*3./(4*!PI))^(1./3.)
   if Keyword_Set(show) then print, Total(sph), esRad, vol/trueVol
   results[*,testNum] = [vol, vol/trueVol, Abs(1.-vol/trueVol), esRad, $
      esRad/sphereRad, Abs(1.-esRad/sphereRad)]
 endfor

 tab = String(9B)
 print, "Category",tab,"Mean",tab,"StDev",tab,"Max",tab,"Min"
 for i=0,N_Elements(headings)-1 do Print, headings[i],tab,Mean(results[i,*]), $
   tab,StdDev(results[i,*]),tab,Max(results[i,*]),tab,Min(results[i,*])
End
```

               IDL code for conducting off-center sphere volume test.

## Appendix C: Blob3D shape calculations

This Appendix briefly describes how 3-D shape calculations are conducted in Blob3D software (Ketcham, 2005; Ketcham and Mote, 2019), as they apply to measuring grain shape for apatite (or any mineral grain for which a shape analysis is conducted).

The measurement process is illustrated in animation 97BS-CR8C.mp4 in the Supplement, which illustrates the shape calculation on several apatite grains in sample 97BS-CR8. The measurement process consists of generating a 3-D shape and measuring the area of its projection (i.e., outline or shadow) over various angles. The procedure first finds the mean projected area by projecting the shape over a uniform distribution of orientations. It then uses the minimum and maximum projected area found in that sampling as starting points to find the true minimum and maximum projected areas via an optimization algorithm (which looks like "jiggling" the shape in the animation). It then calculates the circularity as the ratio of the maximum projection perimeter to a circle with the same area. The routine then finds the longest caliper dimension (ShapeA) or, in other words, the longest dimension that would be measured in 3-D using a caliper. After finding the projection with the longest caliper dimension, the object is rotated around the long axis to find the longest caliber dimension orthogonal to it (ShapeB). The third shape parameter (ShapeC) is the caliper dimension orthogonal to the first two, which is found by rotating the object 90°. Finally, the procedure uses the same method but in the opposite order, finding the shortest caliper dimension (BoxC), the shortest dimension orthogonal to it (BoxB), and the caliper dimension orthogonal to those (BoxA).

The ShapeABC parameters correspond to the long-standing traditional shape measurement method for rounded or irregular particles (Sneed and Folk, 1958; Wilson and Huang, 1979), but the BoxABC parameters (Blott and Pye, 2008) are more appropriate for regular shapes. For example, for a perfect cube, ShapeA is the longest corner-to-corner distance, which will be longer than ShapeB and ShapeC, while BoxA, BoxB, and BoxC will all have the same value: the cube edge length. When measuring an apatite grain, BoxC will usually be the "flattest" part of the hexagonal cross section, BoxB will be the orthogonal corner-to-corner distance of the hexagon, and BoxA will be the length in the prismatic direction unless it is fragmented or has a very low aspect ratio.

**Supplement.** The supplement related to this article is available online at: https://doi.org/10.5194/gchron-1-1-2019-supplement.

**Author contributions.** EHGC collected and processed data, made figures, and contributed to the writing of the paper. RAK initiated the study, processed data, and contributed to the writing of the paper. DFS initiated the study with RAK and contributed to the writing of the paper.

**Competing interests.** The authors declare that they have no conflict of interest.

**Acknowledgements.** We thank Jessie Maisano for acquiring and reconstructing the CT data. These data were collected at the UTCT NSF Multi-User Facility. This paper was improved by helpful reviews from Christoph Glotzbach and two anonymous reviewers.

**Financial support.** This research has been supported by the National Science Foundation, Division of Earth Sciences (grant no. 1762458).

**Review statement.** This paper was edited by Cecile Gautheron and reviewed by Christoph Glotzbach and two anonymous referees. This work was conducted through Jackson School of Geosciences funds to Daniel F. Stockli and an NSF GRF and WHOI postdoc scholarship to Emily H. G. Cooperdock.

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
