# Peer review of "Resolving the effects of 2D versus 3D grain measurements on apatite (U-Th)/He age data and reproducibility"

_Geochronology, 2019_

## Referee Comment (RC1) · Christoph Glotzbach (Referee) · 14 Jun 2019

General comments

Dear Authors, Overall I found the manuscript well written, structured and the topic is of interest for the thermochronological community. The applied analysis is of high quality, but lacks some comments on accuracy and general applications to 'normal' samples (fewer grains). See my Scientific and Technical comments below for details.

Scientific comments 1) I am not totally convinced that the 3D-CT measurements are accurate enough to judge the quality of 2D measurements. The resolution of the voxel

[Figure]

is 4-5 $\mu$m, relatively large compared to a typical grain size of 100 $\mu$m. The authors should convince the reader that the resolution is high enough to use their CT measurments as reference. Maybe you provide some real/synthetic data to prove that the resolution is good enough. You have provided some information in the text, but I am not totally satisfied with that. 2) You have estimated the hexagonal cross-section assuming an equal-sided hexagonal cross section and state correctly that this is not an adequate assumption for all grains. Since it is quite easy to measure the cross section for each grain with a bit more effort, please state why you haven't done this and/or calculate for extreme cases how much uncertainty you add by assuming the equal-sided cross section. 3) You conclude that although estimates of volume, surface area, ESR and Ft are deviating significantly for individual grains they partly average out or in average (for all measured grains) do not deviate from the CT derived values. You have not considered that the usual amount of grains measured in a bedrock sample is around 3 or 4 and in this case the deviations you observed are very likely not cancelling out. You could for instance provide the reader with some estimates of the possible deviation for 3-4 grains by (1) randomly resample 3-4 grains for each sample and (2) calculate the deviation from the complete dataset of different parameters (e.g. ESR, Ft). I hope you find my comments and suggestions helpful.

Technical corrections: Page 2, Line 10-30: No references here, please add relevant references. Page 3, Line 3: Herman et al. (2007) and Glotzbach et al. (2019) did this before. Page 3, Line 5: You do report results from the study of Evans on the accuracy of Ft values, why not reporting results from Glotzbach et al. (2009) on grain measurements? Page 7, Line 6: Not all reader might know what you mean with UFt and ThFt, please explain and probably use U-Ft and Th-Ft which is easier to read. Page 7, Line 7: Some or most of your grains do have tips, why have you not included them in this calculation? There are equations to do that, e.g. Ketcham et al. (2011). Page 7, Line 8: Here you use W for half-width, but further in the equations you use r. Please be consistent. Page 7, Line 18: I would prefer to write ESR = 3 * V/SA Page 7, Line 21: My understanding is that the density of apatite is closer to 3.2 g/cm3, maybe 3.18.

A reference would be nice to have. Page 7, Line 23: Give reference, e.g. Farley 2002. Page 8, Line 10: Please can you add a few examples of how you fit the CT-scanned grains with boxes. Please show examples where this method works fine and if present also some examples that could maybe not fit that easily. Page 9, Line 18: Do you start from the center of the voxel, or somewhere within the randomly chosen voxel? Since the voxel resolution is not that high (4.6-5 microns), starting from the center of the vexel may bias the calculation. Please clarify. Page 9, Line 25: If stopping distance/voxel size > 4 means the resolution is high, the sign should be opposite <!? Can you clarify for which sample/isotope this is true for your dataset and when you have used this super-sampling approach. Page 10, Line 32: Although the result will be quite identical, you have to correct the He content and not the uncorrected age (put the ejected He back in the grain). Page 11, Line 15-18: I would simply omit the grain and you can delete this section. Page 11, Line 29-30 and Page 12, Line 1: How serious is this issue, please report how this can happen and how often and to what degree. You could make repeated measurements on the same grain. Our microscope system is saving the magnification of each picture taken and we do not have yet found any errors. Please also have a look if the grains are somewhat different from other grains (more complex geometry). Page 15, Line 6: Please explain what you mean with simple geological histories Page 15, Line 18: Please explain why you have not used the equations of Ketcham et al. (2011) and did some measurements of the hexagonal cross section, e.g. using double-sided tape. On page 16, Line 20 you indirectly suggest that this should be done. Page 17, Line 10-12: This might be true for the average, but we should still care about the deviation of single grains since deviations might not cancel out if only a few grains are analyses. I would suggest you to randomly (maybe 1000 times) sample 3 or 4 grains from one sample and measure the mean deviation in Ft for those grains. Make a figure with deviation in Ft (x-axis) against probability (y-axis). In this way the results can be better transferred to normal bedrock samples. Page 18, Line 7: Where do you get this from, please provide some evidence for this conclusion (refer to a figure or table). Page 19, Line 7-9: Make sure you pronounce that this is really

only true for these two samples (with fast cooling history). I would expect that even though inclusions may not be always of minerals with high U-Th concentrations they still have a considerable effect on He diffusion. Please mention this to make sure that the reader does not get it wrong and start to pick grains with inclusions. Page 19, Line 14: Please provide the reader with some figure that supports your conclusion. To help the reader to better estimate the importance of broken grains, please report how large the deviation in Ft will be for a range of broken grain scenarios. Maybe you calculate for a few grains of your samples the Ft assuming that the grain broke (1) during mineral separation and (2) broke before cooling. We presented a method to account for this (e.g. Fig. 9 in Glotzbach et al. 2019) and show an example that yield a deviation of 'only' 5% which might be not detectable with your dataset since it is in the same range as other uncertainties. Page 19, Line 29-31: I am not an expert, but I guess you also have to calibrate a CT-scanner or? Page 20, Line 11: The optical microscope along cannot measure eU, please add that you have used an ICP-MS.

Table 1: Can you also report the difference (with sign) not only the absolute of it. Not sure if you really have to report the U Ft and Th Ft, just show the total Ft. Fig. 5: Could you colour-code the relative differences (deviation from the 3D model) and make a small legend in one of the plots?

---

## Referee Comment (RC2) · Anonymous Referee #2 · 17 Jun 2019

General comments: The manuscript "Resolving the effects of 2D versus 3D grain measurements on (U-Th)/ He age data and reproducibility" by Cooperdock et al. provide insight into the issue of accurately measuring apatite grain dimensions for ultimate use in (U-Th)/He age calculations. The manuscript is concise and well written with respect to identifying key problems with conventional 2D microscope approaches and adds to the literature beginning to assess 3D (CT scan) methods for better grain characterization.

Please refer to the attached annotated PDF for in-depth line comments; main points are highlighted below.

[Figure]

I found some references to the general U-Th/He method (early development and application; Zeitler et al. '87, 90s Wolf/Farley/Ehlers/Reiners work etc.) to be lacking and additionally some recent work on age dispersion/analytical approaches for assessing age reproducibility to be uncited, and even prior 3D/CT scan work was cited later in the introduction (Evans et al.; Herman et al.; Glotzbach et al. etc.) - when statements made earlier in manuscript should cite these references as well or perhaps requires reworking of this intro. section.

The authors show that when comparing 2D vs. 3D approaches for age calculations, that there is essentially no difference in the computed U-Th/He ages. While I agree that 3D-CT seemingly yields overall 'better' grain dimension(s)/morphology results, the additional time/cost put forth for CT work is not realistic for most users - i.e. the cost/benefit does not make this approach worthwhile to researchers that do not have easy CT scanner access. Page 18, Lines 20-25 best summarize this conceptually..."Overall, these data suggest that although the 3DFT can provide a more accurate FT correction and varies from 2D estimations by ∼2%, it has a minimal effect on the calculated sample age (1-2%) and no effect on the reproducibility for these two samples. This is not surprising, as a ∼2% error would constitute a negligible proportion of the often cited 6% dispersion derived from analyzing age standards; error propagation indicates that removing a source of 2% error would only reduce an overall 6% error to 5.7%. This points to the importance of other factors in intra-sample dispersion, such as U-Th zonation, and/or excess He from nano-inclusions or high U-Th neighbors." This passage advocates for a more proactive approach to dealing with real and greater sources of age dispersion.

Additionally, this work utilizes apatite grains that were previously known to yield highly reproducible results (and the same holds true after their new work). This was probably the best approach to assess first-order discrepancies between 2D/3D, however, what does a sample look like that yields highly scattered age data? Does 3D help with reducing (some) dispersion? As we know, it is more common for a lot of apatite He ages

to be statistically over-dispersed. I think for additional communal value the authors should characterize some other 'normal' samples.

Major Line-by-line comments (see PDF for all): Page 1, Line 12: How do you quantify eU with 2D microscope or CT information? Page 1, Line 16: Arguably no one uses mass for interpreting age scatter and eU (known to be underestimated)...ESR may be problematic and could be better addressed by the 3D technique. In regards to eU/ESR-age relationships being used in a meaningful way for age scatter interpretation, see recent paper by Fox et al. 2019 G-cubed "Badly behaved detrital U-Th/He ages: problems with He diffusion models or geological models?" Page 1, Line 18: "...effectively no impact on reducing intra-sample age reproducibility" Why would it? We already know what the major causes for intra-sample grain age variation are, even though they are poorly characterized on a routine basis. Even though grain characterization 'errors' have always been a known problem, we know that 2D estimation isn't that bad, otherwise He ages would never work out correctly during FT correction. Well-behaved whole grain samples show this well and you demonstrate this here that for age calcs. it all works out. The magnitude of age dispersion due to inclusions, zoning, or especially 'anomalously sited' 4He far outweigh 2D grain measurement-derived errors for FT correction (see recent papers by Idleman et al. 2018 Chemical Geology or Mcdannell et al., 2018 Geochimica et Cosmochimica Acta –the latter more directly deals with causes of age dispersion) Page 2, Line 11: Why would someone 'assume' parent nuclide heterogeneity without any proof of it? Page 3, Line 23: Number of grains? chi-squared? % dispersion? all informative/useful...because that is a highly precise AFT age. Page 18, Line 15: If it is old compared to the rest of the grains then it must contain anomalous He content. Personally (IMHO), I don't like these types of 'catch-all' statements in papers dealing with unexpected He age data. Rather than the usual retelling of the laundry list of potential issues such as 'He implantation, mineral inclusions, fluid inclusions, U-Th zoning, etc'. - just make a general statement that the age was 'bad or unexpected' - you don't actually know why. If you don't actually have any way of quantitatively assessing the reason then speculation is not useful. By listing

the many issues it subliminally perpetuates the idea that there are so many problems that we don't know where to start or what to do to fix things. Page 19, Line 6: You should stress here that labs should not pick grains with inclusions regardless. It was just done here to illustrate that the CT scan can pick up fluid/mineral inclusions, right? This text as-is makes it seem like it is okay to date as long as they aren't U/Th-bearing inclusions (which you wouldn't know until it was too late).

Please also note the supplement to this comment:
https://www.geochronology-discuss.net/gchron-2019-3/gchron-2019-3-RC2-supplement.pdf

**Supplement:**

[revised manuscript text omitted]

---

## Referee Comment (RC3) · Anonymous Referee #3 · 26 Jun 2019

This paper presents 2D measurement and 3D microCT data for >100 euhedral apatites from two igneous rock samples as well as 83 (U-Th)/He dates for a subset of these grains. It presents a methodology for efficiently acquiring microCT data for a large number of apatites (∼250) with a voxel resolution of 4-5 microns. The authors then compare the volume, surface area, grain mass, ESR values, eU values, and FT corrections derived by the two methods for this apatite suite.

This is a well-written, detailed paper. The primary benefit of this contribution is description of an efficient approach for microCT analysis and Blob 3D data reduction of a large number of grains for improved FT corrections and associated grain geometry,

mass, and concentration information. This will allow others to use this methodology if desired. However, it significantly oversells the quantitative comparison of the 2D measurement and 3D microCT datasets, which should be more appropriately qualified. The deviations between the two sets of measurements presented here are almost certainly minimum differences that likely underestimate those associated with most apatites analyzed in (U-Th)/He labs. It also is unclear the extent to which the 5 um resolution of their microCT approach is an improvement over conventional 2D measurements.

1. The selection of two samples with only euhedral apatites of extremely high quality for this study means that the conclusions regarding the 2D-microCT data comparison are limited to only apatites of this kind. Arguably, such apatites comprise only a small fraction of grains analyzed in (U-Th)/He labs today. The paper casts these conclusions in the title, abstract, introduction, and discussion as being generally applicable. However, they're not. For example, the ∼2% difference in FT factors between the 2D and 3D measurements surely represent minimum uncertainties. For example, it would seem appropriate to insert the following text into this sentence in the abstract: "The data illustrate that the 2D approach...on high-quality euhedral apatites...systematically overestimates grain volumes..." The last sentence of the introduction should be similarly qualified. As should various statements in the discussion. For example, in section 4.2 the authors state that the greatest deviations are caused by user error and not the assumed grain geometry, but this may simply be because the authors only worked on the highest quality apatite subset that most closely approximates the chosen grain geometry. The higher deviations between 2D and microCT results that likely are associated with more typically analyzed apatites might make it more likely for the microCT method described here to be more widely adopted, so I'm surprised that only exceptional apatites were used in this study.

2. The title should indicate that this study is applicable to apatites only – "Resolving the effects of 2D versus 3D grain measurements on apatite (U-Th)/He age data. . ."

3. The 2D-microCT comparison seems to assume no uncertainty on the microCT data

despite the ∼5 um voxel size. Although section 2.4.2 describes various uncertainties associated with the 3D calculations, the bottom line is unclear. Could you please summarize clearly the final uncertainties on the 3D estimates and how this affects the 2D-3D comparison? In the end, how good is a 5 um resolution for determining 3D grain-measurements, especially for apatites on the small end of what is analyzed by (U-Th)/He?

4. The authors seem to dismiss the importance of surface roughness on their results, but detecting it is below their ∼5 um voxel size. Again, I feel that this points again toward the need to qualify some of their generalized statements about uncertainties.

5. In section 4.3 Regarding the discussion of inclusions, I encourage the authors to use more cautious language. As written, non-experts could read their language to mean that inclusions don't matter and picking apatites that contain them would be fine. We know this is not the case. Of course many apatites contain inclusions that aren't U-Th bearing and may not affect the data. The issue is the inability to discriminate between inclusions that are or are not U-Th bearing. Unless there is a way to discriminate, apatites with high-density inclusions shouldn't be analyzed.

6. The second and third paragraphs of the introduction should include appropriate references.

---

## Author Comment (AC1) · 12 Aug 2019

Response to: Interactive comment on "Resolving the effects of 2D versus 3D grain measurements on (U-Th)/ He age data and reproducibility" by Emily H. G. Cooperdock et al. Christoph Glotzbach (Referee) Âăchristoph.glotzbach@uni-tuebingen.de

Please find our responses below each comment. We also attached a revised version of the manuscript and additional appendices with tracked changes to this review.

General comments Dear Authors, Overall I found the manuscript well written, struc-

tured and the topic is of interest for the thermochronological community. The applied analysis is of high quality, but lacks some comments on accuracy and general applications to 'normal' samples (fewer grains). See my Scientific and Technical comments below for details.

Scientific comments

1) I am not totally convinced that the 3D-CT measurements are accurate enough to judge the quality of 2D measurements. The resolution of the voxel is 4-5 m, relatively large compared to a typical grain size of 100 m. The authors should convince the reader that the resolution is high enough to use their CT measurments as reference. Maybe you provide some real/synthetic data to prove that the resolution is good enough. You have provided some information in the text, but I am not totally satisfied with that.

Response: We have added a detailed summary of tests that address these concerns as Appendices B and C, and with Figure B1. In sum, we show that our implementation of measuring grain radii and FT in Blob3D provides a high degree of accuracy and precision even on very small grains at low resolutions where voxels are 20% of the grain radius.

2) You have estimated the hexagonal cross-section assuming an equal-sided hexagonal cross section and state correctly that this is not an adequate assumption for all grains. Since it is quite easy to measure the cross section for each grain with a bit more effort, please state why you haven't done this and/or calculate for extreme cases how much uncertainty you add by assuming the equal-sided cross section.

Response: Point well taken – Although some labs do measure the third dimension, we note that many labs do not routinely do so. As measuring practices vary by lab, we chose to mimic the simplest measuring technique for comparison with the 3D CT data.

3) You conclude that although estimates of volume, surface area, ESR and Ft are

deviating significantly for individual grains they partly average out or in average (for all measured grains) do not deviate from the CT derived values. You have not considered that the usual amount of grains measured in a bedrock sample is around 3 or 4 and in this case the deviations you observed are very likely not cancelling out. You could for instance provide the reader with some estimates of the possible deviation for 3-4 grains by (1) randomly resample 3-4 grains for each sample and (2) calculate the deviation from the complete dataset of different parameters (e.g. ESR, Ft). I hope you find my comments and suggestions helpful.

Response: We have addressed this suggestion by taking the average of four random grains from each sample 1000 times for Mean Ft and Age. The results of this are added to Figure 7, Table 2, and discussed in the text in sections 4.1.4 and 4.2.

Technical corrections: Page 2, Line 10-30: No references here, please add relevant references.

Response: References have been added.

Page 3, Line 3: Herman et al. (2007) and Glotzbach et al. (2019) did this before.

Response: Herman and Glotzbach are cited on this sentence now.

Page 3, Line 5: You do report results from the study of Evans on the accuracy of Ft values, why not reporting results from Glotzbach et al. (2009) on grain measurements?

Response: A sentence is added that reports the Glotzbach results on Ft.

Page 7, Line 6: Not all reader might know what you mean with UFt and ThFt, please explain and probably use U-Ft and Th-Ft which is easier to read.

Response: UFt and ThFt is referenced to Farley et al., 1996, which has an explanation of the derivations of the equations. We tell the reader to reference this in section 2.4.1.

Page 7, Line 7: Some or most of your grains do have tips, why have you not included them in this calculation? There are equations to do that, e.g. Ketcham et al. (2011).

Response: We purposefully chose to compare the 'simplest' 2D measurement approach with the 3D data, with the idea that it will provide an important point of comparison as many labs use this approach.

Page 7, Line 8: Here you use W for half-width, but further in the equations you use r. Please be consistent.

Response: This has been changed to "r".

Page 7, Line 18: I would prefer to write ESR = 3 * V/SA

Response: This is changed.

Page 7, Line 21: My understanding is that the density of apatite is closer to 3.2 g/cm3, maybe 3.18. A reference would be nice to have.

Response: This is a typo and now has been fixed to 3.2.

Page 7, Line 23: Give reference, e.g. Farley 2002.

Response: This has been added.

Page 8, Line 10: Please can you add a few examples of how you fit the CT-scanned grains with boxes. Please show examples where this method works fine and if present also some examples that could maybe not fit that easily.

Response: We have added as Appendix C a discussion and animation that demonstrates the way Blob3D fits boxes.

Page 9, Line 18: Do you start from the center of the voxel, or somewhere within the randomly chosen voxel? Since the voxel resolution is not that high (4.6-5 microns), starting from the center of the vexel may bias the calculation. Please clarify.

Response: We've clarified the text to state that we first randomize which voxel, and then randomize the location within that voxel.

Page 9, Line 25: If stopping distance/voxel size > 4 means the resolution is high, the

sign should be opposite <!? Can you clarify for which sample/isotope this is true for your dataset and when you have used this supersampling approach.

Response: Sign switched. We've also clarified that we use the 238U stopping distance for this test. Appendix B describes super-sampling in more detail.

Page 10, Line 32: Although the result will be quite identical, you have to correct the He content and not the uncorrected age (put the ejected He back in the grain).

Response: We have noted this comment, but decided to leave the text as-is, e.g., Farley et al., 1996.

Page 11, Line 15-18: I would simply omit the grain and you can delete this section.

Response: We chose to leave this grain as part of the discussion of the introduction of user-error. We added some text to say how the importance of user error will vary by lab, microscope, software, etc.

Page 11, Line 29-30 and Page 12, Line 1: How serious is this issue, please report how this can happen and how often and to what degree. You could make repeated measurements on the same grain. Our microscope system is saving the magnification of each picture taken and we do not have yet found any errors. Please also have a look if the grains are somewhat different from other grains (more complex geometry).

Response: This issue is going to vary widely in different labs due to the variety of microscopes and software used. In this case, it was an older software that did not record the magnification. Text has been added to qualify that the issue of 'user error' will vary widely by lab. We bring it up because it ended up being an important source of error in our study, and an issue that needs to be addressed lab-by-lab.

Page 15, Line 6: Please explain what you mean with simple geological histories

Response: The geologic histories are explained in the Background section. By this we mean that they come from crystalline samples that experiences relatively fast exhuma-

tion and no subsequent known reheating. This sentence has been revised.

Page 15, Line 18: Please explain why you have not used the equations of Ketcham et al. (2011) and did some measurements of the hexagonal cross section, e.g. using double-sided tape. On page 16, Line 20 you indirectly suggest that this should be done.

Response: We recognize that different labs have different protocols for grain measurements on the microscope. We chose to measure the grains this way because it is commonly practiced and has the greatest amount of assumption built into it, thereby we expect to see the largest possible deviations

Page 17, Line 10-12: This might be true for the average, but we should still care about the deviation of single grains since deviations might not cancel out if only a few grains are analyses. I would suggest you to randomly (maybe 1000 times) sample 3 or 4 grains from one sample and measure the mean deviation in Ft for those grains. Make a figure with deviation in Ft (x-axis) against probability (y-axis). In this way the results can be better transferred to normal bedrock samples.

Response: We have added this analysis of the data to Figure 7 and discussed it in the text in sections 4.1.4 and 4.2. In short, our 1000-test trial showed that the overall population results with 1-sigma uncertainty we reported match that of the 4-grain trial runs well.

Page 18, Line 7: Where do you get this from, please provide some evidence for this conclusion (refer to a figure or table).

Response: We have added Table and Figure references to this section.

Page 19, Line 7-9: Make sure you pronounce that this is really only true for these two samples (with fast cooling history). I would expect that even though inclusions may not be always of minerals with high U-Th concentrations they still have a considerable effect on He diffusion. Please mention this to make sure that the reader does not get it wrong and start to pick grains with inclusions.

Response: We've changed the sentence to make it more direct that inclusions can be (and often are) a source for dispersion, just not in these samples apparently.

Page 19, Line 14: Please provide the reader with some figure that supports your conclusion. To help the reader to better estimate the importance of broken grains, please report how large the deviation in Ft will be for a range of broken grain scenarios. Maybe you calculate for a few grains of your samples the Ft assuming that the grain broke (1) during mineral separation and (2) broke before cooling. We presented a method to account for this (e.g. Fig. 9 in Glotzbach et al. 2019) and show an example that yield a deviation of 'only' 5% which might be not detectable with your dataset since it is in the same range as other uncertainties.

Response: We do not include a plot of the grain habits vs age because we thought it wasn't more informative than the information provided in Table 2. This study was not designed to assess the effects of broken grains, as few grains within our samples exhibited clear 'broken' ends. This is also supported by the relatively high reproducibility of aliquot ages, and we do not note any specific trend with 'older' ages and broken-looking surfaces. In any event, our software is not yet in a state where it can attempt to "complete" broken grains, which is a difficult problem when habits vary greatly. As such, we do not assess the effect of broken grains and Ft corrections in detail.

Page 19, Line 29-31: I am not an expert, but I guess you also have to calibrate a CT-scanner or?

Response: Modern CT scanners should not need to be calibrated, but we have added a paragraph that discusses user inputs and best practices and pitfalls.

Page 20, Line 11: The optical microscope along cannot measure eU, please add that you have used an ICP-MS.

Response: Yes, eU has been removed from that sentence.

Table 1: Can you also report the difference (with sign) not only the absolute of it. Not

[Figure]

sure if you really have to report the U Ft and Th Ft, just show the total Ft.

Response: The positive or negative difference is shown in the 3D/2D fractions, where 3D/2D < 1 is equivalent to a negative difference, and 3D/2D > 1 is related to a positive difference. Then the absolute percent is reported to highlight the magnitude of difference.

Fig. 5: Could you colour-code the relative differences (deviation from the 3D model) and make a small legend in one of the plots?

Response: We revised Figure 5 by adding contour lines to the graphs that shows the % difference.

Please also note the supplement to this comment:
https://www.geochronology-discuss.net/gchron-2019-3/gchron-2019-3-AC1-supplement.pdf

**Supplement:**

[revised manuscript text omitted]

---

## Author Comment (AC2) · 12 Aug 2019

Response to: Interactive comment on "Resolving the effects of 2D versus 3D grain measurements on (U-Th)/ He age data and reproducibility" by Emily H. G. Cooperdock et al. Anonymous Referee #2

Please find our responses below each comment. We also attached the revised manuscript and appendices with tracked changes.

General comments: The manuscript "Resolving the effects of 2D versus 3D grain measurements on (U-Th)/ He age data and reproducibility" by Cooperdock et al. provide

insight into the issue of accurately measuring apatite grain dimensions for ultimate use in (U-Th)/He age calculations. The manuscript is concise and well written with respect to identifying key problems with conventional 2D microscope approaches and adds to the literature beginning to assess 3D (CT scan) methods for better grain characterization. Please refer to the attached annotated PDF for in-depth line comments; main points are highlighted below.

I found some references to the general U-Th/He method (early development and application; Zeitler et al. '87, 90s Wolf/Farley/Ehlers/Reiners work etc.) to be lacking and additionally some recent work on age dispersion/analytical approaches for assessing age reproducibility to be uncited, and even prior 3D/CT scan work was cited later in the introduction (Evans et al.; Herman et al.; Glotzbach et al. etc.) - when statements made earlier in manuscript should cite these references as well or perhaps requires reworking of this intro. section.

Response: References have been added to the introduction, including earlier reference to previous CT work.

The authors show that when comparing 2D vs. 3D approaches for age calculations, that there is essentially no difference in the computed U-Th/He ages. While I agree that 3D-CT seemingly yields overall 'better' grain dimension(s)/morphology results, the additional time/cost put forth for CT work is not realistic for most users - i.e. the cost/benefit does not make this approach worthwhile to researchers that do not have easy CT scanner access.

Response: We do see the point of this statement, and we've made a conscious effort not to over-sell the technique (as endorsed by the reviewer in his next sentence). However, as costs continue to fall and capabilities continue to rise, and new uses are thought of and developed, we believe the utilization of this technology will continue to grow.

Page 18, Lines 20-25 best summarize this conceptually..."Overall, these data suggest

that although the 3DFT can provide a more accurate FT correction and varies from 2D estimations by âĹij2%, it has a minimal effect on the calculated sample age (1-2%) and no effect on the reproducibility for these two samples. This is not surprising, as a âĹij2% error would constitute a negligible proportion of the often cited 6% dispersion derived from analyzing age standards; error propagation indicates that removing a source of 2% error would only reduce an overall 6% error to 5.7%. This points to the importance of other factors in intra-sample dispersion, such as U-Th zonation, and/or excess He from nano-inclusions or high U-Th neighbors." This passage advocates for a more proactive approach to dealing with real and greater sources of age dispersion.

Response: We agree with this statement.

Additionally, this work utilizes apatite grains that were previously known to yield highly reproducible results (and the same holds true after their new work). This was probably the best approach to assess first-order discrepancies between 2D/3D, however, what does a sample look like that yields highly scattered age data? Does 3D help with reducing (some) dispersion? As we know, it is more common for a lot of apatite He ages to be statistically over-dispersed. I think for additional communal value the authors should characterize some other 'normal' samples.

Response: We generally agree with this statement. As the reviewer states, we purposefully selected reproducible and high quality apatite for this study as a "best case scenario" assessment. We have also now reported the results when we randomly subsample 4 grains from our dataset, to make for a more realistic test. We ran this simulation 1000 times and report the results in Figure 7. These results show more variation, and generally slightly better reproducibility in the 3D method, though still within error of the 2D, and within error of the whole population statistics. However, we do not agree that characterizing more and more-dispersed samples would be helpful; our two samples already feature differing degrees of age dispersion, and we show the FT component of dispersion is similar. Adding in more excess dispersion that we by definition don't understand will not provide further clarity.

Major Line-by-line comments (see PDF for all): Page 1, Line 12: How do you quantify eU with 2D microscope or CT information?

Response: We removed eU from this sentence to avoid confusion. eU quantification is described in Appendix A and Methods. The eU is calculated based on the U, Th and Sm concentrations using Eq. (7). The concentration data are determined by isotope dilution via the ICP-MS (see methods) and using the mass of the grain, which is calculated by the grain dimensions. Therefore, the grain measurement technique indirectly feeds into the eU concentration through the calculation of the grain mass.

Page 1, Line 16: Arguably no one uses mass for interpreting age scatter and eU (known to be underestimated)...ESR may be problematic and could be better addressed by the 3D technique. In regards to eU/ESR-age relationships being used in a meaningful way for age scatter interpretation, see recent paper by Fox et al. 2019 G-cubed "Badly behaved detrital U-Th/He ages: problems with He diffusion models or geological models?"

Response: We mention eU (which is a function of grain mass) here in reference to the effect of radiation damage on He diffusivity, which is still used for inverse modeling and age-scatter interpretation. We rephrase the sentence a bit to clarify this. We also add a reference to the Fox paper in the discussion.

Page 1, Line 18: "...effectively no impact on reducing intra-sample age reproducibility" Why would it? We already know what the major causes for intra-sample grain age variation are, even though they are poorly characterized on a routine basis. Even though grain characterization 'errors' have always been a known problem, we know that 2D estimation isn't that bad, otherwise He ages would never work out correctly during FT correction. Well-behaved whole grain samples show this well and you demonstrate this here that for age calcs. it all works out. The magnitude of age dispersion due to inclusions, zoning, or especially 'anomalously sited' 4He far outweigh 2D grain measurement-derived errors for FT correction (see recent papers by Idleman et al.

[Figure]

2018 Chemical Geology or Mcdannell et al., 2018 Geochimica et Cosmochimica Acta
–the latter more directly deals with causes of age dispersion)

Response: We motivate this study in the last paragraph of the introduction (Pg 3 Line
11). Although the impact of grain measurements on He age errors is 'known' to not be
'that bad', there hasn't been a quantitative systematic evaluation of just how much error
it contributes to the He age of a sample. Evans et al 2008 and Glotzbach et al 2019
are the only other studies that compare 2D and 3D grain measurements. Our study
builds on this previous work in several ways: 1) previous work has used a of max 24
grains, whereas we use > 100 for a more statically robust analysis. 2) No previous work
actually dated the grains after characterizing them. Here we collect the aliquot ages
and calculate the mean sample ages for the entire population and by subsampling the
data (the latter added upon revision). 3) We show that a 5 $\mu$m resolution is sufficient
for CT scanning characterization, which enables many more grains to be measured at
a time (100's). 4) We present a CT mounting and scanning method + new software
to efficiently and cost-effectively collect 3D measurement data for (U-Th)/He. 5) We
highlight that 2D measurement error can have a much larger effect on eU and ESR,
which are ultimately often necessary for making sense of dispersed (U-Th)/He data.

Page 2, Line 11: Why would someone 'assume' parent nuclide heterogeneity without
any proof of it?

Response: We say this because most (U-Th)/He measurements are done on whole
grains without complimentary measurements to check for parent nuclide zoning. In
these cases, the corrections + (U-Th)/He age are calculated with the assumption of
homogeneous parent nuclide distribution.

Page 3, Line 23: Number of grains? chi-squared? % dispersion? all informa-
tive/useful...because that is a highly precise AFT age.

Response: Surpless et al., 2002 report a P(X2) = 75.4%, and measured 25 grains and
100 track lengths. Text to this effect has been added to the manuscript. Full analysis

information are of course also available in the original papers.

Page 18, Line 15: If it is old compared to the rest of the grains then it must contain anomalous He content. Personally (IMHO), I don't like these types of 'catch-all' statements in papers dealing with unexpected He age data. Rather than the usual retelling of the laundry list of potential issues such as 'He implantation, mineral inclusions, fluid inclusions, U-Th zoning, etc'. - just make a general statement that the age was 'bad or unexpected' - you don't actually know why. If you don't actually have any way of quantitatively assessing the reason then speculation is not useful. By listing the many issues it subliminally perpetuates the idea that there are so many problems that we don't know where to start or what to do to fix things. Page 19, Line 6: You should stress here that labs should not pick grains with inclusions regardless. It was just done here to illustrate that the CT scan can pick up fluid/mineral inclusions, right? This text as-is makes it seem like it is okay to date as long as they aren't U/Th-bearing inclusions (which you wouldn't know until it was too late).

Response: We write that implanted He "potentially" contributed to this outlier, but that the reason isn't evident from our data. We've removed the 'potentially due to He implantation' part.

Please also note the supplement to this comment: https://www.geochronology-discuss.net/gchron-2019-3/gchron-2019-3-RC2- supplement.pdf

Response: We have gone through the supplement and made edits according to these suggestions as well.

Please also note the supplement to this comment:
https://www.geochronology-discuss.net/gchron-2019-3/gchron-2019-3-AC2-
supplement.pdf

---

## Author Comment (AC3) · 12 Aug 2019

Response to:
Please find our response to each comment below. We have also attached a revised version of the manuscript with appendices showing tracked changes.

This paper presents 2D measurement and 3D microCT data for >100 euhedral ap-

atites from two igneous rock samples as well as 83 (U-Th)/He dates for a subset of these grains. It presents a methodology for efficiently acquiring microCT data for a large number of apatites (âĹij250) with a voxel resolution of 4-5 microns. The authors then compare the volume, surface area, grain mass, ESR values, eU values, and FT corrections derived by the two methods for this apatite suite.

This is a well-written, detailed paper. The primary benefit of this contribution is description of an efficient approach for microCT analysis and Blob 3D data reduction of a large number of grains for improved FT corrections and associated grain geometry, mass, and concentration information. This will allow others to use this methodology if desired. However, it significantly oversells the quantitative comparison of the 2D measurement and 3D microCT datasets, which should be more appropriately qualified. The deviations between the two sets of measurements presented here are almost certainly minimum differences that likely underestimate those associated with most apatites analyzed in (U-Th)/He labs. It also is unclear the extent to which the 5 um resolution of their microCT approach is an improvement over conventional 2D measurements.

1. The selection of two samples with only euhedral apatites of extremely high quality for this study means that the conclusions regarding the 2D-microCT data comparison are limited to only apatites of this kind. Arguably, such apatites comprise only a small fraction of grains analyzed in (U-Th)/He labs today. The paper casts these conclusions in the title, abstract, introduction, and discussion as being generally applicable. However, they're not. For example, the âĹij2% difference in FT factors between the 2D and 3D measurements surely represent minimum uncertainties. For example, it would seem appropriate to insert the following text into this sentence in the abstract: "The data illustrate that the 2D approach...on high-quality euhedral apatites...systematically overestimates grain volumes..." The last sentence of the introduction should be similarly qualified. As should various statements in the discussion. For example, in section 4.2 the authors state that the greatest deviations are caused by user error and not the assumed grain geometry, but this may simply be because the authors only worked

on the highest quality apatite subset that most closely approximates the chosen grain geometry. The higher deviations between 2D and microCT results that likely are associated with more typically analyzed apatites might make it more likely for the microCT method described here to be more widely adopted, so I'm surprised that only exceptional apatites were used in this study. Response: We purposefully chose high quality apatite with fast cooling histories in order to target the effect of 2D vs 3D measurement techniques on final age calculations. In order to assess the error introduced by assumed grain geometries, we sought out reproducible samples so we could remove other factors that contribute to age dispersion. Although others have presented 3D CT data for apatite before, the previous comparison studies did not date the grains, and they were done on ≤ 24 grains. Here we present data for >100 grains for a more statistically robust comparison and follow it through with dating to compare final age dispersion and results.

Response: We take the point that these results are minimum estimates, and have edited the text to acknowledge this in the Introduction and the Discussion. We also sub-sampled our data to assess the deviation between 2D and 3DCT methods for a sample size of 4 aliquots – which more typically reflects the number of grains analyzed for a single sample. The results were consistent with the statistics presented for the entire population, which is further evidence that these error estimates are robust for apatite with a similar history. We also want to note that although these two samples had abundant apatite, which were 'good quality', we sampled a variety of grain morphologies, sizes, and with inclusions.

2. The title should indicate that this study is applicable to apatites only – "Resolving the effects of 2D versus 3D grain measurements on apatite (U-Th)/He age data. . ."

Response: We've changed the title.

3. The 2D-microCT comparison seems to assume no uncertainty on the microCT data despite the âĹij5 um voxel size. Although section 2.4.2 describes various uncertainties associated with the 3D calculations, the bottom line is unclear. Could you please summarize clearly the final uncertainties on the 3D estimates and how this affects the 2D-3D comparison? In the end, how good is a 5 um resolution for determining 3D grain-measurements, especially for apatites on the small end of what is analyzed by (U-Th)/He?

Response: We have addressed this in detail in the newly added Appendices B and C, and we show that the 4-5 micron voxel resolution is able to produce precise and accurate grain radii, even when the voxel size is 20% of the grain radius.

4. The authors seem to dismiss the importance of surface roughness on their results, but detecting it is below their âĹij5 um voxel size. Again, I feel that this points again toward the need to qualify some of their generalized statements about uncertainties.

Response: We acknowledge that surface roughness, and relatedly, CT smoothness, is a challenging issue to quantify in 2.4.2. However, we do not believe it is important at all for FT correction, because the alpha stopping process, both in reality and as simulated, is essentially a $\sim$20-$\mu$m smoothing filter (except for Sm, of course, but there we are talking about a percent of a percent in terms of effect). We added text to clarify this point.

5. In section 4.3 Regarding the discussion of inclusions, I encourage the authors to use more cautious language. As written, non-experts could read their language to mean that inclusions don't matter and picking apatites that contain them would be fine. We know this is not the case. Of course many apatites contain inclusions that aren't U-Th bearing and may not affect the data. The issue is the inability to discriminate between inclusions that are or are not U-Th bearing. Unless there is a way to discriminate, apatites with high-density inclusions shouldn't be analyzed.

Response: We have edited this sentence to more clearly state that inclusions should be avoided.

6. The second and third paragraphs of the introduction should include appropriate references.

Response: References have been added to these paragraphs.

Please also note the supplement to this comment:
https://www.geochronology-discuss.net/gchron-2019-3/gchron-2019-3-AC3-supplement.pdf

[Figure]

**Supplement:**

[revised manuscript text omitted]